# Waxing-and-Waning: a Generic Similarity-based Framework for Efficient Self-Supervised Learning

**Sheng Li[1], Chao Wu[3], Ao Li[4], Yanzhi Wang[3], Xulong Tang[1], Geng Yuan[2]**
[1]University of Pittsburgh   [2]University of Georgia   [3]Northeastern University
[4]University of Arizona
{shl188,xulongtang}@pitt.edu       geng.yuan@uga.edu
{cha.wu,yanz.wang}@northeastern.edu       aoli1@arizona.edu

## Abstract

Deep Neural Networks (DNNs), essential for diverse applications such as visual recognition and eldercare, often require a large amount of labeled data for training, making widespread deployment of DNNs a challenging task. Self-supervised learning (SSL) emerges as a promising approach, which leverages inherent patterns within data through diverse augmentations to train models without explicit labels. However, while SSL has shown notable advancements in accuracy, its high computation costs remain a daunting impediment, particularly for resource-constrained platforms. To address this problem, we introduce SimWnW, a similarity-based efficient self-supervised learning framework. By strategically removing less important regions in augmented images and feature maps, SimWnW not only reduces computation costs but also eliminates irrelevant features that might slow down the learning process, thereby accelerating model convergence. The experimental results show that SimWnW effectively reduces the amount of computation costs in self-supervised model training without compromising accuracy. Specifically, SimWnW yields up to 54% and 51% computation savings in training from scratch and transfer learning tasks, respectively.

## 1 Introduction

The conventional supervised learning paradigm that underpins their training process relies heavily on large amounts of labeled data (Li & Feng, 2019; Li et al., 2019; Hou et al., 2022). To address these limitations, researchers increasingly turn to Self-Supervised Learning (SSL) as a promising alternative and have made significant progress over the last few years (Bachman et al., 2019; Chen et al., 2020a; Zheng et al., 2021; Xu et al., 2022; Cao et al., 2023; Li et al., 2024). SSL intends to learn the inherent representations that are invariant under different distortions by maximizing the similarity of representations obtained from training samples using different augmentation methods (Misra & Maaten, 2020; Reed et al., 2021; Zbontar et al., 2021).

While recent work has come a long way in terms of improving the accuracy of SSL (He et al., 2020; Ren et al., 2022; Zhu et al., 2023), the exorbitant training cost associated with SSL methods is still a critical challenge of SSL. For example, SSL designs like BYOL (Grill et al., 2020) demand 23 times the computation costs, attributed to $8.8\times$ more required training iterations and $2.6\times$ higher computation cost per iteration (Wen & Li, 2021) compared to the supervised learning counterparts. This will inevitably impede the actual deployment of SSL in real-world applications, especially for resource-limited platforms such as edge devices.

Although a few recent works have begun to acknowledge the importance of the efficiency of SSL, this area is still severely neglected by the community. Recent work (Addepalli et al., 2022) introduces a rotation prediction task along with the original training process to improve the convergence speed of SSL. And Meng et al. (2022) introduces a contrastive dual gating mechanism with extra conditional paths to learn the dynamic gating policy and reduce the computation FLOPs. The existing methods often require increased complexity on top of the original SSL paradigm to improve

efficiency. So, it is natural to raise a question: *Is there a more general and effective method that can significantly improve the training efficiency of SSL?*

Considering the training paradigm of the SSL that leverages different data augmentations on two branches with a siamese encoder model used, it results in a unique property of SSL compared to the conventional supervised learning methods (Tian et al., 2020; Chen et al., 2020a). That is, the augmented input images and feature maps on the two branches inherently have a certain degree of similarity. This is a natural opportunity that could be potentially used for computation saving or simplifying. However, it is insufficient to lead us directly to a simple solution. It is not clear whether similar and dissimilar regions of augmented input images and feature maps in the two branches are equivalently crucial for SSL and whether the similarity remains invariant in low-level features and high-level features. And how can we effectively utilize the similarity to improve training efficiency?

Motivated by these questions, we make a comprehensive exploration of the impact of similar regions on SSL accuracy. And we explore two types of methods (i.e., reuse and remove) to exploit the similarities for computation-saving. We find that eliminating the computation on similar regions of augmented input images and each layer's activations can significantly reduce the computation and speed up SSL. To mitigate the region shrinking problem caused by convolution layers, we propose a strategy to effectively and efficiently identify and expand high-similarity regions to ensure a decent overall computation saving. This strategy can be considered a waxing-and-waning process. Putting it all together, we propose our SIMWNW, a generic and efficient SSL framework that can significantly reduce training costs and improve the convergence speed of SSL.

Our SIMWNW framework is generic and can be easily applied to different SSL training methods for training cost-saving. We evaluate our framework in both training from scratch and transfer learning tasks and validate the effectiveness and generalizability of SIMWNW. Specifically, in training from scratch tasks, compared to representative SSL works, SIMWNW provides significant computation savings, peaking at 54% and averaging at 40%, and without sacrificing accuracy. In transfer learning tasks, SIMWNW shows a notable reduction in computation costs, peaking at 51% and averaging at 48%, without accuracy loss. We also compare SIMWNW to efficient SSL approaches. In both training from scratch and transfer learning tasks, SIMWNW consistently outperforms SOTA works, achieving an average computational cost reduction of 18% and 14%, respectively.

## 2 BACKGROUND AND RELATED WORK

### 2.1 REPRESENTATIVE SELF-SUPERVISED LEARNING FRAMEWORKS

SimCLR (Chen et al., 2020a) stands out as a pivotal framework in contrastive self-supervised learning. It operates by maximizing the similarity between representations of differently augmented versions of the same image and concurrently reducing the similarity with representations of other images in the batch. These representations, derived from encoders, capture the essential features of the images. Specifically, for any given image, its representations $\mathbf{z}_i$ and $\mathbf{z}_j$ are treated as a positive pair. The contrastive loss is:

$$\mathcal{L}_{\text{SimCLR}} = -\log \frac{\exp(\mathbf{z}_i \cdot \mathbf{z}_j / \tau)}{\sum_{k=1}^{N} \exp(\mathbf{z}_i \cdot \mathbf{z}_k / \tau)} \tag{1}$$

Here, $\tau$ denotes a temperature parameter that regulates the sharpness of the distribution, $\mathbf{z}_k$ is the representation of other images (negative samples), and $N$ is the batch size.

MoCo (Momentum Contrast) (He et al., 2020) extends the idea of contrastive learning by incorporating a momentum-based encoder and a dynamic dictionary of data samples. This dynamic dictionary facilitates a rich source of negative samples, enhancing the contrastive task. The loss employed in MoCo is akin to SimCLR, though enriched by the abundance of negatives from the dictionary. To stabilize learning and enhance the quality of the dictionary representations, MoCo updates its encoder's parameters using a momentum mechanism. Given two sets of encoder parameters, $\theta_{\text{old}}$ and $\theta_{\text{new}}$, and the momentum coefficient $m$, the update rule is:

$$\theta_{\text{old}} \leftarrow m\theta_{\text{old}} + (1 - m)\theta_{\text{new}} \tag{2}$$

BYOL (Bootstrap Your Own Latent) (Grill et al., 2020) and SimSiam (Chen & He, 2021) stand out for their innovative emphasis on aligning representations from distinct augmentations of a single

image, eliminating the need for negative samples commonly used in other contrastive learning methods. Within BYOL and SimSiam, the concepts of online and target branches are crucial. The online branch processes one augmented view of an image through its neural network encoder, generating a representation in the feature space. Simultaneously, the target branch processes another distinct augmented view of the same image. The primary objective is to align the representation from the online branch to mirror closely that of the target branch. Crucially, back-propagation occurs in the online branch, updating the network weights. This design ensures the alignment of representations between the two branches, aiming to maximize their similarity within the feature space.

## 2.2 EFFICIENT SELF-SUPERVISED LEARNING.

The recent work (Addepalli et al., 2022) argues that a key reason for the slow convergence of self-supervised learning is attributed to the presence of noise in the training objective, and proposes to add the rotation prediction as a noise-free auxiliary training objective to accelerate the convergence speed. Another work (Koçyiğit et al., 2023) increases the training efficiency of self-supervised training by accelerating model convergence through a novel learning rate schedule and an input image resolution schedule. It also proposes a new augmentation method to enhance the quality of augmented images to further accelerate model convergence. Compared to these methods, our SIMWNW focus on exploring and exploiting the similarity inside of input images and intermediate feature maps to save computation costs. Our SIMWNW is compatible with these methods as well.

## 3 MOTIVATION AND EXPLORATION

In SSL, the siamese encoder models are usually used in the two branches, and each branch processes an augmented input derived from the same source image. So, an inherent similarity exists between the augmented images processed in the two branches. Based on this unique property of SSL, we have an assumption that regions with high similarity between the images in the two branches might have limited contribution to the self-supervised model training. If this is true, we have the potential to reduce computation costs. In this section, we first investigate the importance of regions with varying degrees of similarity in the two augmented images from the same source image. Following this, we explore two strategies — reuse and remove — to reduce computation in regions perceived to be less important. We further illustrate the advantages and drawbacks associated with each method.

### 3.1 IMPORTANCE OF DIFFERENT REGIONS ON AUGMENTED IMAGES

To verify our assumption, we conduct a preliminary study focusing on the importance of similar versus dissimilar regions in augmented images during training. Using the SimSiam framework as our baseline (Figure 1a), each image is transformed into two distinct views (i.e., augmented images), serving as inputs of two branches in the SimSiam framework. In our study, we preprocess the two augmented images before sending them to be processed by the model. We first divide the augmented image of the online branch into blocks of the same size. Each block undergoes a block-matching process (exhaustive search) to locate its most similar counterpart in the paired augmented image of the target branch. Here, the similarity between blocks is quantified using the PSNR (peak signal-to-noise ratio) (Hore & Ziou, 2010). An example of block matching is shown in Figure 1b. For the

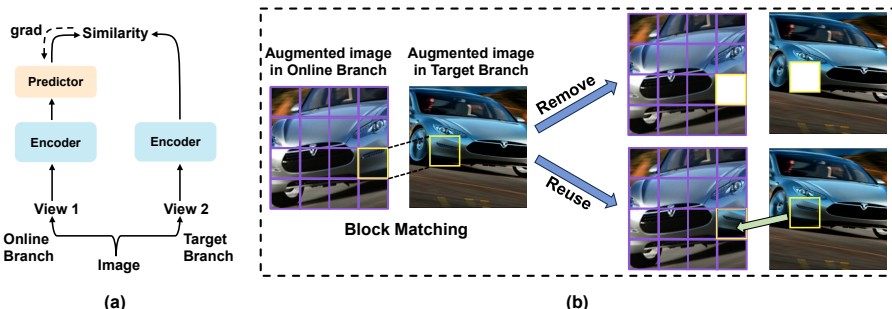

Figure 1: (a) SSL framework. (b) Examples of block matching, reuse, and remove.

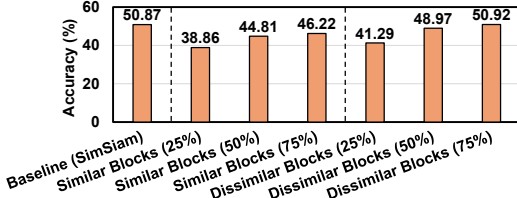 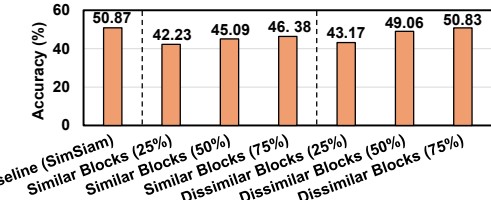

(a) Removed a portion of the similar/dissimilar block pairs within the two augmented images. The notation ($x\%$) represents that $x\%$ of the most similar/dissimilar block pairs are retained, implying that $(1-x)\%$ of the block pairs have been removed.

(b) Reuse the computation results of a portion of the similar/dissimilar blocks in the online branch by using the convolution results of their corresponding blocks in the target branch. The ($x\%$) indicates that $x\%$ of the most similar/dissimilar blocks are not "reused".

Figure 2: Preliminary study. The encoder is ResNet50, which is pre-trained on the ImageNet dataset. Then the ResNet50 is trained for 100 epochs using Stanford Cars dataset. The accuracy results are obtained by the linear evaluation method.

image block marked with yellow boxes in the online branch, the most similar image block found in the target branch is also marked with yellow boxes.

Once the search process is done, we create many block pairs in the augmented images of two branches. We search for the most similar counterpart in the target branch for each image block in the online branch, trying to find the corresponding block pair that is semantically related. We then sort the block pairs by their similarity (their similarity is already calculated during the search process) and remove either the most similar or dissimilar pairs. In this case, if we remove block pairs with high similarity, we can evaluate the effect of dissimilar regions on training. Conversely, by removing block pairs with low similarity, we can assess the impact of similar regions on training. The modified augmented images then continue through the standard SimSiam processing pipeline.

In this experiment, we use ResNet50 (He et al., 2016) as the encoder and use Stanford Cars dataset (Krause et al., 2013) as an example. The augmented images are resized to dimensions of 224x224, and we opt for a block size of 30x30, drawing from recommendations in image block matching studies (Wei & Wu, 2013; Shin et al., 2008). We will discuss the impact of block size later in Section 5.2. Figure 2a presents the experimental results. Notably, when retaining an equivalent percentage of block pairs, prioritizing the retention of dissimilar ones yields better accuracy. The reason behind this is that the dissimilar regions provide difficulty for learning, compelling the model to learn meaningful representations (Tian et al., 2020). In addition, when the appropriate number of block pairs are retained (i.e., dissimilar blocks (75%) in Figure 2a), there is no accuracy drop. These results highlight the importance of the dissimilar regions in augmented images for self-supervised learning, affirming our assumption and motivating us to reduce computation on similar regions.

## 3.2 Reuse vs. Remove

By leveraging the similarity in SSL, there are two possible ways that we can use to reduce the computation: reuse and remove (depicted in Figure 1b). For the reuse method, we can replace some blocks on the online branch with corresponding similar blocks on the target branch. In this way, we can directly reuse the computation results (i.e., the corresponding blocks on the output feature map of convolutional layers) from the target branch as an approximation of the online branch result, avoiding computing the convolution output of the similar blocks on two branches separately. For the remove method, we directly eliminate specific regions of images, causing the convolution operation to skip these regions.

To compare the effects of reuse and remove, we conduct an experiment following the setup described in Section 3.1 and applied the reuse operation. The results are presented in Figure 2b. Comparing Figures 2a and 2b, we observe that at a high reuse/remove ratio (i.e., retain 25% blocks), the reuse operation delivers higher accuracy as it keeps more semantic information. However, with a moderate level of block retention (50% and 75%), both operations yield comparable accuracies. Notably, when the majority of blocks (i.e., 75% dissimilar blocks) are retained, either the remove or reuse method will not degrade accuracy.

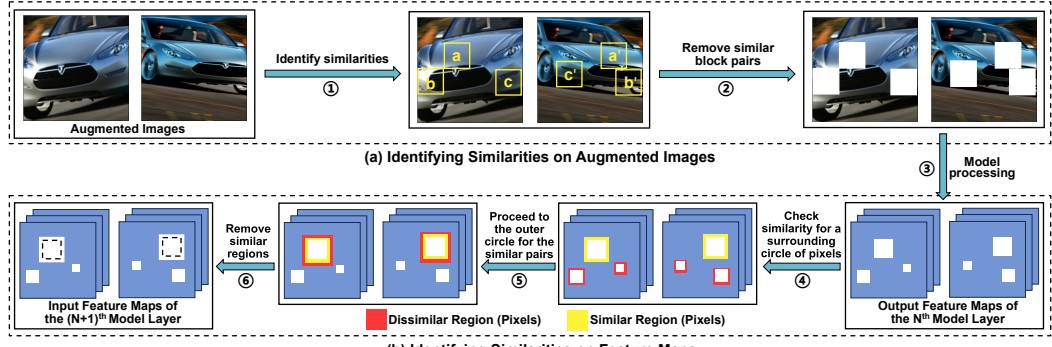

Figure 3: Overview of SIMWNW framework. The similar image blocks are marked with yellow boxes and matched with lettered notations. The removed regions (i.e., pixels) are shown as white boxes. In steps ④ and ⑤, the region with color surrounding the white boxes is the region where we check for similarity. The region inside the dashed boxes after step ⑥ represents the previously removed pixels prior to step ④. The white region outside the dashed boxes denotes newly identified and removed similar regions from the feature map.

Yet, when it comes to computation savings, reuse operation falls far behind remove due to three reasons: i) Reuse operation benefits only one branch. The corresponding augmented images in the other branch still undergo the regular model processing, essentially meaning that computation savings are halved. ii) The benefits of the reuse operation don't translate well during back-propagation. This is because SSL frameworks like MoCo, BYOL, and SimSiam perform back-propagation exclusively in one branch. This singular branch back-propagation renders computation reuse non-viable. iii) Furthermore, frameworks like MoCo and BYOL use models with identical architectures but varying parameters across the two branches, eliminating the possibility of computation reuse even during forward propagation. In contrast, the remove operation fully eliminates the convolution computation in the removed region. In a nutshell, when appropriately strategized, the remove operation is more promising for computational efficiency.

## 4 FRAMEWORK DESIGN

In this section, we propose a generic similarity-based efficient self-supervised learning framework SIMWNW to reduce the computation cost. As shown in Figure 3, we first launch a block-matching process to identify (①) similar blocks on the two augmented images and remove (②) them for computation-saving. However, one challenge is that the removed region in the augmented images tends to progressively diminish and eventually vanish layer by layer due to the inherent properties of convolution. The consequence is that the potential for computation savings gradually decreases as the image is processed deeper into the network layers. To address this, SIMWNW not only removes similar block pairs from the input augmented images but also identifies and removes similar regions in the intermediate feature maps (i.e., activation), which are steps ④ to ⑥ in Figure 3.

### 4.1 IDENTIFYING SIMILARITIES ON AUGMENTED IMAGES

The first step in SIMWNW is to identify the similarities in the two augmented images from the same original image. We divide one of the two augmented images into blocks, and for each block, we search for a similar block in the other paired augmented image. The similarity between image blocks is quantified using the well-established lightweight PSNR metric, which is calculated as:

$$Similarity = 10 \cdot \log_{10}\left(\frac{\text{MAX}_I^2}{\text{MSE}}\right) \tag{3}$$

where $MAX_I$ is the maximum possible pixel value of the image, and $MSE$ is the Mean Squared Error between the two compared images. In Section 3, we apply an unoptimized exhaustive search method, which traverses the entire image using the same block size. However, the exhaustive search is not desirable for training acceleration due to the computation overhead introduced. Moreover,

such a search could possibly overlook the image's semantic information, leading to the removal of two blocks that appear similar but carry different semantic meanings. As shown in Figure 4, blocks $c$ and $c'$ look similar due to the lighting conditions, but they semantically diverge. Removing such blocks could potentially hurt the image's semantic information, decreasing model performance.

To circumvent this, our strategy involves searching regions in the paired augmented image that share semantic links. Instead of scanning the entire image, our focus narrows down to a specific region surrounding a block's counterpart in the paired augmented image. This approach is based on the belief that semantically related blocks in two augmented images are highly possible to come from the same region of the original image. This approach not only reduces computation overhead but also ensures searching semantically related blocks, safeguarding the image's overall semantic essence.

Pinpointing the semantically related corresponding regions necessitates tracking pixel position shifts during augmentations such as random crops or horizontal flips. It's worth noting that our search is not confined strictly to the direct counterpart block. We search for an expanded region (empirically set to four times the block size). This expansion ensures a balance of computational efficiency and precision. Furthermore, our search is not a full sweep of this expanded region. We apply the diamond search algorithm (Zhu & Ma, 2000) for efficient block-matching.

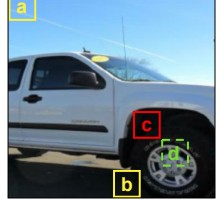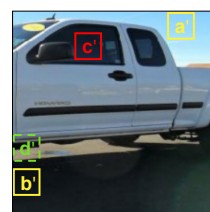

Figure 4: An example of using exhaustive search to match similar blocks. The similar image blocks are marked with yellow and red boxes and matched with lettered notations. Note the regions within the two dotted green boxes are not similar.

Given the widespread use of random crop augmentation in SSL, there will be instances where one image's blocks lack a direct match within its counterpart augmented images. Here, by default, we scan the nearest region in the paired image, even if the exact counterpart is cropped out. At first blush, this might seem at odds with our semantically aware strategy. Yet, in our practice, this method proves adept at discarding voluminous, semantically sparse backgrounds. Take, for instance, the blocks $a$ and $a'$ or $b$ and $b'$ in Figure 4. Even though they aren't direct matches, removing them doesn't detract from the image's semantic information. Conversely, regions brimming with vital details, like intricate objects, remain largely untouched by this method due to their inherent dissimilarity. A case in point is the relationship between blocks $d$ and $d'$ in Figure 4. Despite block $d'$ being closest to where the counterpart block of $d$ would be located, they will not be considered similar.

## 4.2 IDENTIFYING SIMILARITIES ON FEATURE MAPS

### 4.2.1 CHALLENGES: SHRINKING REMOVED REGION SIZE

Identifying and removing similar regions in augmented images offers a pathway to saving computation costs. Yet, this strategy has its limitations: the size of the removed region typically shrinks and eventually disappears as the images proceed through network layers. As a result, the computation saving is progressively reduced as the images proceed deeper layer by layer. This is due to the intrinsic behavior of convolution operations. If a convolution kernel traverses across both removed and normal regions in the augmented images and feature maps, the convolution operations can not be skipped. The convolution operations can be skipped only when the kernel is entirely within the removed region. For a clearer picture, consider a scenario depicted in Figure 5. After undergoing a convolution with a 3x3 kernel, the removed region shrinks from 5x5 to 3x3, ef-

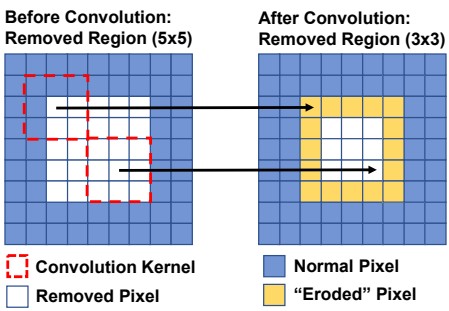

Figure 5: An example to show how the removed region shrinks after convolution operation. The kernel size here is 3x3 and the stride is 1.

fectively "waning" its outermost pixel boundary. This waning phenomenon becomes inevitable when the kernel size is larger than 1x1, which is often the case with CNNs that often use 3x3 or larger kernels. Some networks do not even contain 1x1 kernels, making this issue more pronounced.

This directs our attention to strategizing ways to sustain the size of the removed region. Understanding that various layers in a neural network cater to extracting features at differing levels is key: the front layers are responsible for low-level feature extraction (e.g., detecting edges), whereas the deeper layers extract high-level features (Yosinski et al., 2014; Zeiler & Fergus, 2014; Chen et al., 2023). This layered hierarchy suggests that as images move deeper in the model layers, the emergence of similar regions on the feature maps may become more evident as similarities in high-level features on the feature maps gradually emerge.

### 4.2.2 EXPANDING THE SIZE OF REMOVED REGION

Building on this insight, we propose to identify and remove similar regions in the feature maps of various model layers. Specifically, before the convolution operation on the feature maps, we would scrutinize the surrounding regions around the removed regions within a pair of feature maps. We advocate for a concentric circle approach to identify similarities. That is, we begin by evaluating the similarity of the immediate circle of pixels around the removed region (step ④ in Figure 3). If the immediate circles of pixels in two paired feature maps are similar, we proceed to the next pair of outer circles (step ⑤ in Figure 3). This iterative examination continues until the similarity for a certain circle pair falls below the similarity threshold. In contrast to block-matching in augmented images, identifying similarities in feature maps is a direct one-to-one similarity check without a searching process. For a visual illustration, see the feature maps after step ⑥ in Figure 3. The white regions outside the dashed box in the feature maps represent regions newly identified as similar and consequently removed.

The reason we focus on regions surrounding the removed region arises from the presumption that neighboring regions might exhibit higher similarities. For instance, referring to the upper-left convolution kernel in Figure 5, although its computation is not skipped and its output is an "eroded" pixel, it is crucial to recognize that the source of this eroded pixel contains a portion of the removed (similar) region. Moreover, given that changes in images are usually relatively flat (Huang et al., 2000), it is reasonable to assume the regions around the removed regions are more likely to be similar.

To further refine this process, we advocate for a channel-wise approach to determine similarities within the feature map, followed by a tailored removal procedure for each channel based on its distinct similarity results. Given that feature maps can have a plethora of channels (sometimes even in the hundreds), and the spatial distribution of similar regions can vary significantly among channels, a holistic approach encompassing all channels would not be efficient.

## 5 EVALUATION

We evaluate the proposed SIMWNW framework in both training from scratch and transfer learning tasks. In the training from scratch task, we use three representative datasets CIFAR-10, CIFAR-100 (Krizhevsky & Hinton, 2009), and ImageNet (Deng et al., 2009). In the transfer learning task, the encoder is pre-trained on the ImageNet dataset and then trained on three datasets Stanford Cars, FGVC Aircraft (Maji et al., 2013), and Caltech-UCSD Birds (CUB) (Wah et al., 2011). The batch size in all the experiments is set to 256. For all datasets except CIFAR, we employ ResNet50 as the encoder. For the CIFAR dataset, we use ResNet18 as the encoder. We apply SIMWNW in two representative SSL frameworks for evaluation: SimSiam and SimCLR.

In our experiments, the similarity threshold is set to 20 to consider two image blocks are similar. We select this threshold based on the common practice in wireless image transmission, where a PSNR value of 20 between a compressed image and its original version is generally deemed to be of acceptable similarity for many applications (Muralikrishna et al., 2013; Minallah et al., 2021). However, this similarity threshold can be modified to adapt to the unique needs of specific applications. The overhead introduced by similarity calculation is included in the results and we present a detailed analysis of overhead in Appendix C. When we do the block-matching for augmented images, we use a block size of 30x30 for 224x224 images and scale down to a 5x5 block size for 32x32 images. When the images cannot be evenly divided, we accommodate overlapping blocks at the edges. We follow the linear evaluation protocol (Goyal et al., 2019) in our experiments, where the pre-trained model remains static, and only the appended linear classification layer undergoes fine-tuning. All the results in this paper are the average of 3 runs using different random seeds.

Table 1: Results of training from scratch experiments. The encoder for ImageNet is ResNet50 and the encoder for CIFAR is ResNet18. The model is trained for 800 epochs.

| Method | ImageNet | | | CIFAR-10 | | | CIFAR-100 | | |
|---|---|---|---|---|---|---|---|---|---|
| | Acc. | Training FLOPs | Training Time | Acc. | Training FLOPs | Training Time | Acc. | Training FLOPs | Training Time |
| SimCLR | 66.39 | 100% | 100% | 90.16 | 100% | 100% | 57.34 | 100% | 100% |
| SimCLR + Back Razor | 66.30 | 85% | – | 89.82 | 64% | – | 57.60 | 69% | – |
| SimCLR (same FLOPs as Ours) | 64.76 | 80% | – | 86.07 | 48% | – | 50.46 | 52% | – |
| SimCLR + SIMWNW (Ours) | 66.25 | 80% | 89% | 90.10 | 48% | 68% | 57.69 | 52% | 73% |
| SimSiam | 71.12 | 100% | 100% | 90.80 | 100% | 100% | 57.21 | 100% | 100% |
| SimSiam + Back Razor | 71.04 | 84% | – | 91.05 | 61% | – | 57.87 | 70% | – |
| SimSiam (same FLOPs as Ours) | 69.10 | 81% | – | 87.28 | 46% | – | 52.18 | 53% | – |
| SimSiam + SIMWNW (Ours) | 71.28 | 81% | 90% | 91.17 | 46% | 65% | 57.94 | 53% | 71% |

Table 2: Results of transfer learning experiments. The encoder is ResNet50, which is pre-trained on ImageNet and then trained on the three datasets for 100 epochs using self-supervised learning methods. After that, we apply linear evaluation protocol to get the accuracy results.

| Method | Stanford Cars | | | FGCV Aircraft | | | CUB-200 | | |
|---|---|---|---|---|---|---|---|---|---|
| | Acc. | Training FLOPs | Training Time | Acc. | Training FLOPs | Training Time | Acc. | Training FLOPs | Training Time |
| SimCLR | 46.12 | 100% | 100% | 48.43 | 100% | 100% | 35.78 | 100% | 100% |
| SimCLR + Back Razor | 46.10 | 62% | – | 48.19 | 60% | – | 36.27 | 59% | – |
| SimCLR (same FLOPs as Ours) | 43.03 | 53% | – | 46.74 | 51% | – | 33.07 | 55% | – |
| SimCLR + SIMWNW (Ours) | 46.38 | 53% | 79% | 48.26 | 51% | 75% | 36.15 | 55% | 84% |
| SimSiam | 50.87 | 100% | 100% | 51.82 | 100% | 100% | 38.40 | 100% | 100% |
| SimSiam + Back Razor | 50.83 | 65% | – | 51.68 | 61% | – | 38.07 | 57% | – |
| SimSiam (same FLOPs as Ours) | 46.22 | 50% | – | 48.17 | 55% | – | 36.80 | 49% | – |
| SimSiam + SIMWNW (Ours) | 50.95 | 50% | 78% | 51.76 | 55% | 73% | 38.22 | 49% | 74% |

## 5.1 MAIN RESULTS

**Training from scratch.** Table 1 shows the accuracy and training FLOPs of training from scratch experiments. We apply our proposed SIMWNW framework to two SSL frameworks SimCLR and SimSiam for evaluation. We compare our framework with a recognized activation gradient pruning approach Back Razor (Jiang et al., 2022), which prunes the activation gradient with lower magnitudes to reduce computation costs. In our experiments, we modulate the pruning ratio of Back Razor to ensure a similar accuracy to SIMWNW. As shown in the table, compared to the SimCLR and SimSiam baselines, SIMWNW provides impressive training FLOPs savings, peaking at 54% and averaging at 40%. Importantly, this computational advantage does not come at the expense of accuracy. In comparison with Back Razor, SIMWNW reduces training FLOPs by up to 25%, with an average savings of 18%.

**Transfer learning.** In addition to the training from scratch tasks, we further evaluate the performance of SIMWNW on transfer learning tasks. The experimental results are shown in Table 2. As one can observe, compared to SimCLR and SimSiam baselines, SIMWNW showcases a commendable reduction in computation costs without compromising accuracy, peaking at 51% and averaging at 48%. When compared to Back Razor, SIMWNW manages to reduce computation costs by up to 23%, with an average saving of 14%.

With the savings in training FLOPs, SIMWNW is able to reduce the training time by 24% on average. We also present the accuracy results when SimCLR and SimSiam baselines consume the same training FLOPs with our proposed SIMWNW in Tables 1 and 2, and we can see that SIMWNW consistently provides significantly higher accuracy (3.5% on average) compared to them. The commendable performance of SIMWNW can be attributed to its astute focus on the intrinsic features of augmented images. By eliminating regions of less significance, it not only reduces computation costs but also removes unnecessary noise or irrelevant features that might mislead or slow down the learning process (Wang et al., 2022; Singh et al., 2016). Therefore, it allows the model to learn feature extraction capabilities faster. In other words, the model can converge in fewer epochs, offering further computation savings.

Figure 6 depicts an example of model convergence of training from scratch experiment on the CIFAR-10 dataset. It's evident that applying our SIMWNW framework effectively accelerates the convergence, requiring more than 30% fewer epochs than the SimCLR and SimSiam baselines. The breakdown of computation savings, derived from i) skipping convolution computation for removed regions, and ii) accelerating model convergence, are detailed in Appendix D. We also present a detailed discussion on how the computation savings translate into training time reduction in Appendix E. On the other hand, Back Razor keeps dense activation during forward propagation, which means it could miss opportunities for computation saving.

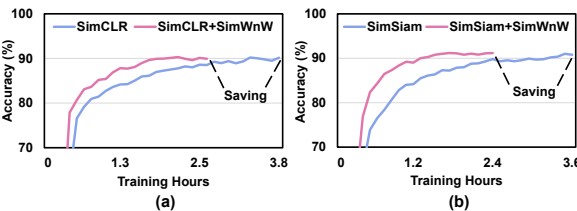

Figure 6: The convergence of ResNet18 on CIFAR-10 dataset of training from scratch experiments.

We also conduct other experiments to evaluate SIMWNW, including comparison and compatibility to some efficient self-supervised learning frameworks (Appendices A.1, A.2, and A.3), experiments on vision transformers (Appendix A.4), and transferability experiments (Appendix A.5).

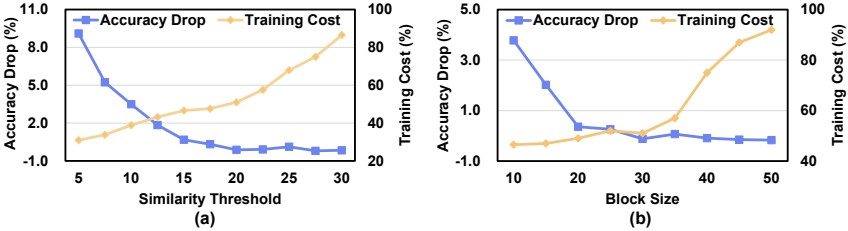

Figure 7: (a) Effect of different similarity thresholds. (b) Effect of different block sizes. All the results are obtained from the transfer learning experiment on the Stanford Cars dataset.

## 5.2 SENSITIVITY STUDY

We investigate the effect of similarity thresholds. Figure 7a shows accuracy drop and training FLOPs (compared to SimCLR and SimSiam baselines) when the similarity threshold varies. As expected, higher thresholds indicate fewer similar block pairs being removed, thus leading to a lower accuracy drop but a higher training computation cost. When the similarity threshold reaches a sufficiently high value (e.g., 20), the accuracy essentially remains stable. However, for applications that are not particularly sensitive to accuracy, opting for an aggressive threshold can save more computation.

We also study the impact of the block size when we conduct block-matching on augmented images. As shown in Figure 7b, a larger block size results in a lower accuracy drop but higher training costs. The reason behind this is that a larger block captures more features and details. As the two augmented images have different transformations applied, the aggregated difference over a larger block will be more pronounced than in a smaller block. Therefore, a larger block size indicates fewer blocks are considered similar. It is worth noting that the optimal block size is tied to the dimension of input images. Users have the flexibility to tailor the block size to suit specific applications. Some other block-matching details and analyses are presented in Appendix B.

## 6 CONCLUSION

In this work, we propose SIMWNW, a similarity-based efficient self-supervised learning framework. By removing the less important similar regions in augmented input images and feature maps, SIMWNW manages to skip unnecessary computations. Moreover, removing the less important regions can also remove irrelevant features that might slow down the learning process. Therefore, SIMWNW is also able to improve the model convergence speed, which can further reduce the computation costs. Our extensive experiments demonstrate that SIMWNW can achieve significant training computation cost savings without compromising accuracy.

ACKNOWLEDGEMENT

The authors would like to thank the anonymous reviewers for their constructive feedback and suggestions. This work is supported in part by NSF grants #2011146, #2154973, #2312157, #2312158, and #2052528.

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

APPENDIX

# A   EXTENDED EXPERIMENTAL RESULTS

## A.1   COMPARISON WITH SOTA EFFICIENT SELF-SUPERVISED LEARNING METHODS

In this section, we compare our SIMWNW with two state-of-the-art efficient self-supervised learning methods (Addepalli et al., 2022; Koçyiğit et al., 2023). Addepalli et al. (2022) identifies the noise present in the training objective as a primary factor contributing to the slow convergence in self-supervised learning. To mitigate this, they introduce rotation prediction as an additional, noise-free training objective, aimed at expediting the model convergence. On the other hand, Koçyiğit et al. (2023) proposes a strategy to accelerate model convergence through a combination of an innovative learning rate schedule and an input image resolution schedule. They also introduce a new image augmentation technique, designed to improve the quality of augmented images, thereby further accelerating the convergence of the model. For these two SOTA methods, we modulate their training epochs to ensure a similar accuracy to SIMWNW for a fair comparison. The experimental result is shown in Table 3. We can observe that SIMWNW consistently provides higher training time reduction with similar model accuracy. The reason behind this is that some computation reduction operations applied by these two methods cannot effectively translate into time reduction. For example, using low-resolution images to reduce the training computation might degrade model performance, resulting in the need for more training epochs.

Table 3: Comparison with SOTA efficient SSL methods.

| Method | ImageNet | | CIFAR-10 | |
|---|---|---|---|---|
| | Accuracy | Training Time | Accuracy | Training Time |
| SimCLR | 66.39 | 100% | 90.16 | 100% |
| SimCLR + (Addepalli et al., 2022) | 66.21 | 91% | 90.04 | 80% |
| SimCLR + (Koçyiğit et al., 2023) | 66.30 | 96% | 90.12 | 78% |
| SimCLR + SIMWNW (Ours) | 66.25 | 89% | 90.10 | 68% |
| SimSiam | 71.12 | 100% | 90.80 | 100% |
| SimSiam + (Addepalli et al., 2022) | 71.20 | 94% | 91.19 | 75% |
| SimSiam + (Koçyiğit et al., 2023) | 71.19 | 95% | 91.11 | 81% |
| SimSiam + SIMWNW (Ours) | 71.28 | 90% | 91.17 | 65% |

## A.2   COMPATIBILITY OF SIMWNW

In this section, we further evaluate the compatibility of SIMWNW with two recent efficient self-supervised learning frameworks Fast-MoCo (Ci et al., 2022) and S3L (Cao & Wu, 2021). These two works are complementary to SIMWNW. Specifically, Fast-MoCo accelerates the training process by adding more positive pairs to regulate the training process, thereby accelerating convergence. S3L accelerates the training process by using smaller-resolution images and a partial backbone network. On the other hand, SIMWNW accelerates the SSL in a different dimension, which removes less important image blocks during training.

We integrate our proposed method SIMWNW into these two frameworks. We follow the setting in their paper and use the MoCo v2 framework (Chen et al., 2020b) as the baseline for S3L and MoCo v3 as the baseline for Fast-MoCo. Specifically, Fast-MoCo divides the input image in the online branch into four patches and then combines their four output embeddings to form six new embeddings, each of which involves two patches. In this case, the number of positive pairs is six times as normal training. Thus, it can get more supervision signals in each iteration and thus achieves promising performance with fewer iterations. For S3L, we follow their original setting for the ImageNet experiment in their paper, which uses 52x52 input images to train the model for 800 epochs and then uses 224x224 input images to train the model for 200 epochs. As shown in Table 4, applying SIMWNW to the Fast-MoCo and S3L can further reduce the training cost by 23% and 30% without accuracy loss, respectively, demonstrating the compatibility of SIMWNW.

Table 4: Compatibility of SimWnWwith SOTA efficient SSL framework. The encoder is ResNet50 and the dataset is ImageNet.

| Method | Accuracy | Training Time |
|---|---|---|
| MoCo v2 | 71.12 | 100% |
| S3L (MoCo v2 based) | 69.96 | 65% |
| S3L + SimWnW | 70.06 | 46% |
| MoCo v3 | 72.28 | 100% |
| Fast-MoCo (MoCo v3 based) | 72.46 | 30% |
| Fast-MoCo + SimWnW | 72.30 | 23% |

## A.3 COMPARISON WITH A SPARSE SELF-SUPERVISED LEARNING APPROACH

There are also some other popular efficient learning methods such as layer freezing (Yuan et al., 2022; Li et al., 2022) and sparse training (Yuan et al., 2021; Ma et al., 2024). For a comprehensive evaluation, we compare our proposed SimWnW framework with a sparse training approach tailored for self-supervised learning: Contrastive Dual Gating (CDG) (Meng et al., 2022). CDG exploits spatial redundancy by using a spatial gating function and skips those less important pixels in the feature maps. To ensure a fair comparison, we modulate the prune ratio of CDG to achieve an accuracy that is comparable to our SimWnW framework. Table 5 shows the experimental results. Under similar accuracy, SimWnW outperforms CDG by saving 12% computation costs, demonstrating the superiority of our proposed SimWnW framework.

Table 5: Comparison with sparse training method CDG (Meng et al., 2022). Accuracies are obtained by linear evaluation. The encoder is ResNet18 with 800 training epochs.

| Method | CIFAR-10 | | CIFAR-100 | |
|---|---|---|---|---|
| | Accuracy | Training FLOPs | Accuracy | Training FLOPs |
| SimCLR | 90.16 | 100% | 57.34 | 100% |
| SimCLR + CDG | 90.01 | 56% | 57.50 | 58% |
| SimCLR + SimWnW | 90.10 | 48% | 57.69 | 52% |

## A.4 EXPERIMENTS ON VISION TRANSFORMERS

In this section, we apply the proposed method SimWnW to vision transformers. There is no removed region shrinking problem in vision transformers, which indeed makes it easier to apply SimWnW to ViT. We also do not need to consider the block size since ViT naturally divides the image into many tokens, typically in the size of 16x16. Therefore, removing similar blocks can be directly achieved by removing similar input tokens, resulting in a reduced input sequence length. We use a well-recognized self-supervised vision transformer learning framework DINO (Caron et al., 2021) as our baseline. As shown in Table 6, SimWnW significantly reduces the training cost by 38% without accuracy loss compared to the DINO baseline. We also compare our approach to an efficient vision transformer training framework EViT. For a fair comparison, we apply EViT to DINO and let it consume the same FLOPs as our approach. As shown in Table 6, SimWnW achieves 0.82% higher accuracy than EViT with the same training computation.

Table 6: Evaluation on Vision Transformers. The encoder is DeiT-S and the dataset is ImageNet.

| Method | Accuracy | Training FLOPs | Training Time |
|---|---|---|---|
| DINO | 58.95 | 100% | 100% |
| DINO + EViT | 58.01 | 57% | 72% |
| DINO + SimWnW | 58.83 | 57% | 72% |

## A.5 TRANSFERABILITY

For a more comprehensive comparison, we conduct an experiment that directly fine-tunes the pre-trained ResNet50 model on the downstream task without self-supervised learning on them. In this experiment, SIMWNW is applied to SimCLR and SimSiam baselines during the self-supervised pre-training on ImageNet. Then we directly train the classifier of the pre-trained ResNet50 using three downstream datasets. As shown in Table 7, applying SIMWNW during the pre-training stage will not hurt the accuracy when directly transferring the pre-trained model to downstream datasets.

Table 7: Accuracy results of transferability experiment.

| Method | Stanford Cars | FGCV Aircraft | CUB-200 |
|---|---|---|---|
| SimCLR | 38.49 | 40.46 | 30.10 |
| SimCLR + SIMWNW (Ours) | 39.90 | 41.28 | 30.19 |
| SimSiam | 39.78 | 38.37 | 32.89 |
| SimSiam + SIMWNW (Ours) | 40.46 | 38.92 | 33.66 |

## B BLOCK MATCHING DETAILS

**Search region.** During the block-matching process, for each block, when searching for the most similar block in the paired augmented image, the search region narrows down to a specific region surrounding the block's counterpart. The reason why we search for a specific region is to handle the effects of image scaling and aspect ratio changes. When we perform data augmentation, RandomResizedCrop is a commonly used method, specifically $transforms.RandomResizedCrop(224, scale = (s1, s2), ratio = (r1, r2))$. This method operates as follows. Initially, it randomly selects a scale ratio between s1 and s2, and the image is scaled up or down according to the randomly picked ratio. Additionally, it will randomly pick an aspect ratio between r1 and r2, and the aspect ratio of the image is accordingly modified. After the scaling and aspect ratio adjustment, the image is then cropped and resized, in this case, 224x224 pixels.

When we apply the augmentation to the original image twice to get two augmented images for two branches, the randomly picked scale ratio and aspect ratio are likely to be different for the two augmented images. Therefore, the corresponding regions in the two augmented images, which come from the same region in the original image, can be different in size and shape. Therefore, it is hard to directly pair the exact matches and we need to search for a neighborhood region of the counterpart block. In practice, the image scaling ratio is generally not more than twice, so the length and width of the search region are set to twice the length and width of the block size to ensure that the most similar blocks are included. For example, the search region is 60x60 for a 30x30 block. We also perform a sensitivity study on the search region size and the results in Table 8 show that SIMWNW is robust to different sizes of the search region.

**Block removal ratio.** Table 9 presents the block removal ratio with the default similarity threshold 20 in our experiments. In practice, the block removal ratio varies across datasets. This is because different datasets have different image complexity and different resolutions, so the number of similar blocks that can be found under the same threshold varies.

Table 8: Accuracy when using different search region sizes. The block size is 30x30 and the results are obtained from transfer learning experiments on the Stanford Cars dataset. The base framework is SimSiam.

| Size of Search Region | 45x45 | 60x60 | 75x75 |
|---|---|---|---|
| Accuracy | 50.72 | 50.95 | 50.98 |

## C OVERHEAD ANALYSIS

The computational overhead in our framework primarily arises from the calculation of PSNR, which is employed for similarity checks between a pair of images and feature maps. However, the silver lin-

Table 9: Image block removal ratio.

| Dataset | ImageNet | CIFAR-10 | CIFAR-100 | Stanford Cars | FGCV Aircraft | CUB-200 |
|---------|----------|----------|-----------|---------------|---------------|---------|
| Removal Ratio | 20% | 30% | 30% | 33% | 31% | 35% |

ing is that PSNR computation is relatively lightweight and hence does not introduce a large amount of computation overhead. The formula for PSNR is derived from the Mean Squared Error (MSE) between the two images:

$$\text{MSE} = \frac{1}{MN} \sum_{i=0}^{M-1} \sum_{j=0}^{N-1} [I(i,j) - K(i,j)]^2 \tag{4}$$

where $I$ and $K$ represents two images (blocks). Using the MSE, the PSNR is calculated as $\text{PSNR} = 10 \cdot \log_{10}\left(\text{MAX}_I^2/\text{MSE}\right)$, where $\text{MAX}_I$ is the maximum possible pixel value of the image.

Considering calculating PSNR for two $30 \times 30$ image blocks, the calculation requires $30 \times 30 = 900$ subtractions to determine differences, 900 multiplications to square these differences, and 899 additions to sum the squared differences. While the PSNR computation does encompass additional operations, such as logarithmic operation, their quantity is minimal and they hold negligible numerical impact on computation cost. Overall, the computation amounts to approximately 2,700 FLOPs.

Although the cost for calculating PSNR once is small, we also apply further methods to reduce the overhead by reducing the number of PSNR calculations. For block-matching in the paired augmented images, each block's search space is limited to the semantically related, constrained region in its counterpart image, leading to a substantial reduction in the search space. Within this limited space, the efficient search method diamond search is invoked, enhancing efficiency. Note, when checking the similarity between paired feature maps, the process is a direct one-to-one match without a searching process (the detail of identifying similarities in feature maps is in Section 4.2).

In contrast, performing convolution on two $30 \times 30$ images with a $3 \times 3$ filter, leading to an output space of 28x28 (due to boundary effects), has a much higher computation cost. Each pixel in the image necessitates $9C$ multiplications and $8C$ additions, where $C$ represents the number of output channels. For the entire 30x30 block, the required FLOPs are given by:

$$\text{FLOPs} = 28 \times 28 \times (9C + 8C) \tag{5}$$

Suppose the number of output channels $C = 64$, the convolution operation demands 852,992 FLOPs for a single image. Consequently, for the paired augmented image (blocks) in two branches, the computation reaches 1.7 MFLOPs.

To summarize, a convolution operation on the image blocks requires computational resources that are several orders of magnitude higher than what's required for PSNR calculation on the image blocks. Moreover, our design limits the search space of each block, thereby ensuring that the PSNR is not calculated too many times. As a result, the overhead introduced is minimal when compared to the gains in computation savings. In our experiments, the computation overhead of PSNR is less than 1% and the overhead is already included in the reported results.

## D  COMPUTATION SVAINGS BREAKDOWN

In this section, we break down the computation savings achieved by SIMWNW framework. Through strategically removing regions of less importance from augmented images and feature maps, SIMWNW achieves computational efficiency in two ways: i) skipping convolution computation for the removed regions, and ii) improving the model convergence speed. Table 10 and Table 11 present the breakdown of computation savings in training from scratch and transfer learning tasks, respectively.

## E  ANALYSIS ON FLOPS REDUCTION

The FLOPs reduction of our SIMWNW mainly comes from two aspects. For the first aspect, our SIMWNW can improve the model convergency speed, indicating a fewer number of training itera-

Table 10: Breakdown of Computation Savings in training from scratch experiments. "Skipped Convolution" represents the savings achieved by skipping convolution computation for the removed region. "Epochs Reduction" indicates the savings in training epochs derived from improved model convergence speed.

| Method | ImageNet | | | CIFAR-10 | | | CIFAR-100 | | |
|---|---|---|---|---|---|---|---|---|---|
| | Overall Savings | Skipped Convolution | Epochs Reduction | Overall Savings | Skipped Convolution | Epochs Reduction | Overall Savings | Skipped Convolution | Epochs Reduction |
| SimCLR + SimWnW | 20% | 12% | 11% | 52% | 29% | 32% | 48% | 29% | 27% |
| SimSiam + SimWnW | 19% | 12% | 10% | 54% | 30% | 35% | 47% | 26% | 29% |

Table 11: Breakdown of Computation Savings in transfer learning experiments. "Skipped Convolution" represents the savings achieved by skipping convolution computation for the removed region. "Epochs Reduction" indicates the savings in training epochs derived from improved model convergence speed.

| Method | Stanford Cars | | | FGCV Aircraft | | | CUB-200 | | |
|---|---|---|---|---|---|---|---|---|---|
| | Overall Savings | Skipped Convolution | Epochs Reduction | Overall Savings | Skipped Convolution | Epochs Reduction | Overall Savings | Skipped Convolution | Epochs Reduction |
| SimCLR + SimWnW | 47% | 33% | 21% | 49% | 32% | 25% | 45% | 35% | 17% |
| SimSiam + SimWnW | 50% | 36% | 22% | 45% | 25% | 27% | 51% | 34% | 26% |

tions/epochs required to achieve a target accuracy. This can directly lead to FLOPs reduction and training time savings without any dedicated sparse computation support. Specifically, SimWnW removes the less important regions, resulting in removing irrelevant features that slow down the learning process, thereby improving model convergence speed.

The second aspect of FLOPs reduction is achieved by removing similar blocks. For the ViT-based models, removing similar blocks can be directly achieved by removing similar input tokens, resulting in a reduced input sequence length. This can also directly achieve acceleration, while does not require any dedicated sparse computation support.

For the case of using CNN models, SimWnW indeed requires some support for sparse computation. This is a similar problem faced by the designs in other fields, such as sparse training or weight pruning. This usually can be solved in different ways. For general-purpose devices such as GPUs or mobile devices, SimWnW can be supported by using sparse computation libraries and compiler optimizations. For FPGA platforms, the convolution kernels need to be divided into tiles and computed separately. So, the tiling size used in FPGAs can be aligned with the block size used in the SimWnW. In this way, we can easily skip the computation clock cycle for the corresponding block, leading to direct time-saving. It is worth mentioning that, in our SimWnW, we remove the entire similar blocks during the computation, which creates a coarse-grained sparsity. Compared to the unstructured or irregular sparsity which is usually used in sparse training or weight pruning works, the coarse-grained sparsity created in our SimWnW is much more friendly for sparse computation acceleration on both general-purpose devices and FPGA platforms.

