# OpenReview forum: "Waxing-and-Waning: a Generic Similarity-based Framework for Efficient Self-Supervised Learning"
_ICLR.cc/2024/Conference — ICLR 2024 poster_

### Official Review · Reviewer_wGaZ · 2023-10-28

**Soundness:** 3 good
**Presentation:** 2 fair
**Contribution:** 2 fair
**Rating:** 6
**Confidence:** 5

**Summary:**

This paper proposes an efficient SSL approach called SimWnW. Through studying the impact of similar and dissimilar image regions on SSL performance, the authors find that similar regions are less important and removing them in augmented images (and in feature maps) can significantly reduce the computation cost and improve model convergence. To remove similar regions, the authors propose a new method under the ResNet/ConvNet settings. Specifically, a waxing-and-waning process is proposed for region removal while mitigating the region shrinking problem in convolutional layers. Experiments show that SimWnW can reduce the computation cost of SSL without compromising accuracy -- SimWnW yields up to 54% and 51% computation savings in training from scratch and transfer learning tasks, respectively.

**Strengths:**

- The paper offers a comprehensive exploration of the impact of similar/dissimilar regions on SSL accuracy, which lays a good foundation for a region removal-based method to improve SSL efficiency.
- Strong results in efficiency boost are achieved for two representative SSL frameworks.
- Decent analysis is provided for region removal-related hyper-parameters like similarity threshold and block size.

**Weaknesses:**

- The key hyper-parameter of block removing portion is unspecified, and convincing explanations are missing (see questions below).
- The comparisons with recent related works seem insufficient, e.g. (Addepalli et al., 2022) and (Koc¸yigit et al., 2023).
- The proposed waxing-and-waning method is customed too much to ConvNets. It seems hard to translate to transformers and hence transformer-based SOTA SSL methods (this makes the paper title a bit overclaim).

**Questions:**

Key question around the portion of block removal:
- Intuitively, comparing similar blocks won't generate too much useful signal for SSL. This is validated by Fig. 2 where the performance of "Similar Blocks (x\%)" is consistently worse than "Dissimilar Blocks (x\%)". On the other hand, comparing dissimilar blocks (after removing similar ones), despite being more useful, has a key hyper-parameter of the removing portion (1-x)\% which can significantly affect the learning quality. Specifically, if we remove too much, comparing those top dissimilar blocks either makes learning too hard or the dissimilar blocks may not even be semantically related (which hurts SSL quality). If we gradually increase x\%, the retained blocks would include both dissimilar and relatively similar blocks, which makes the learning signals more balanced for SSL.
- Fig. 2 shows that SSL performance peaks at "Dissimilar Blocks (75\%)". What's the actually used x\% after region removal in SimWnW? If it's 75\% or higher, then it shouldn't lead to that much of computation saving. Fig. 7(a) shows some hint about x in terms of similarity threshold. 1) When the default threshold is set to 20, what's the corresponding x\%? 2) With the default similarity threshold 20, the SSL performance remains about the same but the training cost is increasing. So again, the computation saving is still concerning. Any comments?
- One side question, why the compute saving on ImageNet is much smaller than CIFAR 10/100? This suggests the amount of removed blocks from high-resolution ImageNet images is smaller than that of low-resolution CIFAR images, given the same similarity threshold (if that's how it works). Any intuitions about why this is the case?

Other minor questions:
- To find similar blocks, what's the neighborhood size for searching? Does it depend on augmentation parameters? - since how we crop/rotate/flip images will impact the block locations a lot.
- For "block matching" in pixel space, is PSNR an accurate enough metric? What if the found correspondence is wrong and how well can SimWnW tolerate such errors?

---

> ### Author Response · Authors · 2023-11-21
> **Author Response to Reviewer wGaZ (Part 1/4)**
>
> **We appreciate the valuable comments from the reviewer. We carefully address all the reviewer’s questions and revise the paper accordingly. We hope our response can help alleviate the reviewer's concern.**
>
> ---
>
> ### **Q1. Comparisons with recent related works, e.g. (Addepalli et al., 2022) and (Koc¸yigit et al., 2023).**
>
> Thanks for your suggestion.
>
> We have added the experiments to compare SimWnW with (Addepalli et al., 2022) and (Koc¸yigit et al., 2023). The experimental result is shown in Table R.9. We can observe that SimWnW consistently provides higher training time reduction with similar accuracy.
>
> Specifically, (Addepalli et al., 2022) identify the noise present in the training objective as a primary factor contributing to the slow convergence in self-supervised learning. To mitigate this, they introduce rotation prediction as an additional, noise-free training objective, aimed at expediting the model convergence. On the other hand, (Koc¸yigit et al., 2023) propose a strategy to accelerate model convergence through a combination of an innovative learning rate schedule and an input image resolution schedule. They also introduce a new image augmentation technique, designed to improve the quality of augmented images, thereby further accelerating the convergence of the model.
>
> For these two methods, we modulate their training epochs to ensure a similar accuracy to SimWnW for a fair comparison. The reason why SimWnW outperforms them is that some computation reduction operations applied by these two methods cannot effectively translate into time reduction. For example, using low-resolution images to reduce the training computation might degrade model performance, resulting in the need for more training epochs. We have added the results in Section 5.2 and Table 3 in the revised paper.
>
> We would also like to point out that our approach and these two methods optimize SSL in different dimension and are complementary. (Addepalli et al., 2022) accelerates the learning process by adding additional rotation prediction tasks; and (Koc¸yigit et al., 2023) accelerates the convergence by adopting a novel learning rate schedule, using smaller input image resolution and novel image augmentation methods. On the other hand, our approach focuses on removing less important regions of the image, which indeed operates in the intra-image level. We also want to mention that we have provided the experimental results on combining our approach with two other SLL methods [1][2] in Section 5.4.
>
> >**Table R.9: Comparison with SOTA efficient SSL methods.**
> | **Method**              	| **ImageNet** |       	| **CIFAR-10** |       	|
> |----------------------------------------------|----------------------------------|---------------|----------------------------------|---------------|
> |                                          	| Acc.                         	| Training Time | Acc.                             | Training Time |
> | SimCLR                                   	| 66.39                        	| 100\%     	| 90.16                        	| 100\%     	|
> | SimCLR + (Addepalli et al., 2022)    	| 66.21                        	| 91\%      	| 90.04                        	| 80\%      	|
> | SimCLR + (Koc¸yigit et al., 2023)  | 66.30                        	| 96\%      	| 90.12                        	| 78\%      	|
> | **SimCLR + SimWnW (Ours)**   	| **66.25**                        	| **89\%**      	| **90.10**                        	| **68\%**      	|
> | SimSiam                                  	| 71.12                        	| 100\%     	| 90.80                        	| 100\%     	|
> | SimSiam + (Addepalli et al., 2022)   	| 71.20                        	| 94\%      	| 91.19                        	| 75\%      	|
> | SimSiam + (Koc¸yigit et al., 2023) | 71.19                        	| 95\%      	| 91.11                        	| 81\%      	|
> | **SimSiam + SimWnW (Ours)**  	| **71.28**                        	| **90\%**      	| **91.17**                        	| **65\%**      	|
>
> [1] Fast-MoCo: Boost Momentum-based Contrastive Learning with Combinatorial Patches. ECCV2022.
>
> [2] Rethinking Self-Supervised Learning: Small is Beautiful. arXiv 2103.13559.

---

> ### Author Response · Authors · 2023-11-21
> **Author Response to Reviewer wGaZ (Part 2/4)**
>
> ### **Q2. Adapting SimWnW to vision transformers.**
>
>
> Thank you for your valuable suggestion.
>
> We would like to clarify that it is applicable to apply SimWnW to ViTs. First, there is indeed no removed region shrinking problem in ViTs, which makes it easier to apply SimWnW to the ViTs. Second, we also do not need to consider the block size since ViTs naturally divide the image into many tokens, typically in the size of 16x16.
>
> Therefore, removing similar blocks can be directly achieved by removing similar input tokens, resulting in a reduced input sequence length.
>
> We use a well-recognized self-supervised vision transformer learning framework DINO [1] as our baseline. As shown in Table R.10, SimWnW significantly reduces the training cost by 38\% without accuracy loss compared to the DINO baseline.
>
> We also compare our approach to an efficient vision transformer training framework EViT [2]. For a fair comparison, we apply EViT to DINO and let it consume the same FLOPs as our approach. As shown in Table R.10, SimWnW achieves 0.82\% higher accuracy than EViT with the same training computation.
>
>
> >**Table R.10: Evaluation on Vision Transformers. The encoder is DeiT-S and the dataset is ImageNet.**
> | **Method**                 	| **Acc.** | **Training FLOPs** | **Training Time** |
> |--------------------------------|----------|--------------------|-------------------|
> | DINO                       	| 58.95\%  | 100\%          	| 100\%         	|
> | DINO + EViT                	| 58.01\%  | 57\%           	| 72\%          	|
> | **DINO + SimWnW**  | **58.83\%**  | **57\%**           	| **72\%**          	|
>
>
> We have added the experiments on ViTs in Appendix A and Table 5 in the revised paper.
>
> [1] Caron, Mathilde, et al. "Emerging properties in self-supervised vision transformers." Proceedings of the IEEE/CVF international conference on computer vision. 2021.
>
> [2] Liang, Youwei, et al. "Not all patches are what you need: Expediting Vision Transformers via Token Reorganizations." International Conference on Learning Representations. 2022.
>
>
> ---
>
>
> #### **Q3. Intuitively, comparing similar blocks won't generate too much useful signal for SSL. This is validated by Fig. 2 where the performance of "Similar Blocks (x%)" is consistently worse than "Dissimilar Blocks (x%)". On the other hand, comparing dissimilar blocks (after removing similar ones), despite being more useful, has a key hyper-parameter of the removing portion (1-x)% which can significantly affect the learning quality. Specifically, if we remove too much, comparing those top dissimilar blocks either makes learning too hard or the dissimilar blocks may not even be semantically related (which hurts SSL quality). If we gradually increase x%, the retained blocks would include both dissimilar and relatively similar blocks, which makes the learning signals more balanced for SSL.**
>
> Thank you for the feedback.
>
> Yes, we agree with you that removing too many similar block pairs will affect the performance of the model. The removing ratio is a hyper-parameter, determined by the similarity threshold. We have provided a sensitivity study on the threshold in the paper in Section 5.3.
>
> Removing fewer image blocks does help to maintain the model's accuracy. However, on the other hand, for applications that are not particularly sensitive to accuracy, opting for an aggressive threshold can save more computation.
>
> We added more discussion in Section 5.3 in our revised paper.

---

> ### Author Response · Authors · 2023-11-21
> **Author Response to Reviewer wGaZ (Part 3/4)**
>
> ### **Q4. Fig. 2 shows that SSL performance peaks at "Dissimilar Blocks (75%)". What's the actually used x% after region removal in SimWnW? If it's 75% or higher, then it shouldn't lead to that much of computation saving. Fig. 7(a) shows some hint about x in terms of similarity threshold. 1) When the default threshold is set to 20, what's the corresponding x%?**
>
> Thank you for your insightful question.
>
> We have added the block removal ratio with the default similarity threshold 20 in our experiments, shown in Table R.11. We could see that in most experiments the block removal ratio is larger than 25%. In other words, the x% is lower than 75%.
>
> In practice, the block removal ratio varies across datasets. This is because different datasets have different image complexity and different resolutions, so the number of similar blocks that can be found under the same threshold varies.
> We have added these results to Appendix B and Table 6.
>
> In terms of computation savings, we also want to clarify that the computation saving not only comes from the skipped computation of the removed region but also comes from the faster model convergence since SimWnW removes unnecessary noise or irrelevant features that might slow down the learning process, as stated in Section 5.1 in the paper.
>
>
> >**Table R.11: Image block removal ratio.**
> | **Dataset**   | **ImageNet** | **CIFAR-10** | **CIFAR-100** | **Stanford Cars** | **FGCV Aircraft** | **CUB-200** |
> |---------------|--------------|--------------|---------------|-------------------|-------------------|-------------|
> | **Removal Ratio** | 20\%     	| 30\%     	| 30\%      	| 33\%          	| 31\%          	| 35\%    	|
>
>
> ---
>
>
> ### **Q5. Fig. 7(a) shows some hint about x in terms of similarity threshold. With the default similarity threshold 20, the SSL performance remains about the same but the training cost is increasing. So again, the computation saving is still concerning. Any comments?**
>
> Thanks for your valuable feedback.
>
> I respectfully suppose you are referring to Figure 7(a), which shows that when the similarity threshold is higher than 20, the model performance remains the same but the training cost keeps increasing.
>
> When we increase the similarity threshold, it will result in a fewer number of blocks can meet the similarity requirements, indicating fewer blocks can be removed (since we only remove the blocks that are highly similar). This will reduce the computation saving. Basically, the similarity shrehold controls the trade-off between computation saving and accuracy.
> On the other hand, the accuracy will not increase as it reaches the upper bound model accuracy. It means that in the benchmark in Figure 7(a), setting the threshold as 20 is the choice that can maximize the training cost reduction without affecting the model accuracy (let’s call it the sweet point of the similarity threshold).
>
> We would like to point out that we may not be able to find the optimal sweet point of similarity threshold in all applications. However, despite this, SimWnW still can effectively reduce a large amount of computation without affecting the accuracy, as long as the threshold is within a reasonable range.
>
> In summary, our proposed method does significantly reduce the training FLOPs by skipping the computation on the removed region and accelerating model convergence as removing similar blocks can reduce the complexity of data and let the model focus on more important signals. We also add the results of training time to the results in Tables 1, 2, 3, 4, and 5 in the revised paper to further justify our proposed method.
>
>
> ---
>
> ### **Q6. Why the compute saving on ImageNet is much smaller than CIFAR 10/100? This suggests the number of removed blocks from high-resolution ImageNet images is smaller than that of low-resolution CIFAR images, given the same similarity threshold (if that's how it works). Any intuitions about why this is the case?**
>
> Thanks for your insightful question.
>
> The reasons why the number of removed blocks from high-resolution ImageNet images is smaller than that of low-resolution CIFAR images are two-fold.
>
> First, we apply the block size of 30x30 to the 224x224 images (ImageNet) when performing block-matching. For CIFAR, whose image size is 32x32, we proportionally scale down the block size to 5x5 when doing block-matching. In this case, a larger block used in ImageNet captures more features and details. As the two augmented images have different transformations applied, the aggregated difference over a larger block will be more pronounced than in a smaller block. Therefore, fewer block pairs will be considered similar given the same threshold.
>
> Second, the images of ImageNet are more complex and contain more details compared to CIFAR, and therefore it is more difficult to find similarities.

---

> ### Author Response · Authors · 2023-11-21
> **Author Response to Reviewer wGaZ (Part 4/4)**
>
> ### **Q7. To find similar blocks, what's the neighborhood size for searching? Does it depend on augmentation parameters? - since how we crop/rotate/flip images will impact the block locations a lot.**
>
> Thank you for the questions.
>
> We empirically set the neighborhood size to four times the block size. For example, the search space is 60x60 for a 30x30 block. And your understanding is correct, the neighborhood search region size that we pick does take the augmentation method into consideration.
>
> Intuitively, the most similar block could be at the corresponding location after augmentation such as flip or rotate. However, if we take the random scaling and cropping augmentation (zoom-in/zoom-out) into account (which is very critical in contrastive learning), it will cause some trouble. We may no longer be able to find an exact one-to-one correspondence block pair between the online branch and the target branch. This is one of the reasons that we want to search for the most similar block in a small region.
> Another reason that we search in a small region (instead of exhaustive search) is to reduce the computation overhead and ensure the semantic consistency of the paired blocks.
>
> In practice, the image scaling ratio is generally not more than twice, so the length and width of the search region are set to twice the length and width of the block size (e.g., 60x60 search space for 30x30 blocks) to ensure that the most similar blocks are included.
>
> For better illustration, we also include Table R.12  here, to show the impact of the search region size on accuracy. We have added the explanation and results to Appendix B Table 7 in the revised paper.
>
> >**Table R.12: Accuracy when using different search region sizes. The block size is 30x30 and the results are obtained from the transfer learning experiment on the Stanford Cars dataset. The base framework is SimSiam.**
> | **Size of Search Region** | 45x45 | 60x60 | 75x75 |
> |---------------------------|-------|-------|-------|
> | **Accuracy**          	| 50.72 | 50.95 | 50.98 |
>
>
> ---
>
> ### **Q8. For "block matching" in pixel space, is PSNR an accurate enough metric? What if the found correspondence is wrong and how well can SimWnW tolerate such errors?**
>
> Thanks for your valuable question.
>
> We agree that PSNR might not be the optimal choice and always pinpoint accurate correspondences. However, based on our extensive experimental results, including: different datasets (e.g., CIFAR10, CIFAR100, ImageNet, Cars, Aircraft, CUB-200), different network structures (e.g., ResNet18, ResNet50, Deit), and different tasks (SSL training from scratch, SSL transfer learning), it suggests that our proposed approach can effectively reduce computation without sacrificing accuracy using PSNR as the metric.
>
> There is an interesting thing that we would like to mention. As we claimed in section 5, we choose PSNR=20 in our work as the threshold to consider two image blocks are similar. Because, for the common practice in wireless image transmission, using a PSNR value of 20 between a compressed image and its original version is generally considered to have acceptable similarity for many applications. And as proved by our experiments (Fig.7), using a PSNR threshold of 20 for block similarity is also a desirable choice for self-supervised learning, which does not degrade the accuracy while achieving decent computation reduction. This could be considered as a guide for threshold selection for future research and exploring other possible metrics is also a direction that is worth further study.
>
> We added more discussion in Appendix E in our revised paper.

---

> ### Author Response · Authors · 2023-11-22
> **Author Response to Reviewer wGaZ**
>
> Dear Reviewer wGaZ,
>
> Thanks for your time and reviewing efforts! We appreciate your thorough comments.
>
> We provide suggested results in the authors' response, including more experiments and explanations on the block removal details, experimental results on vision transformers, and comparisons with SOTA efficient self-supervised learning framework.
>
> We hope our responses have answered your questions. It would be our great pleasure if you would consider updating your review or score.
>
> Best,
>
> Authors

---

> > ### Comment · Reviewer_wGaZ · 2023-11-23
> > **Response to Author Feedback**
> >
> > Thanks for your detailed feedback. Most of my concerns have been addressed, and I think it's important to integrate the new results and discussions into the manuscript. I've raised score to 6.

---

> > > ### Author Response · Authors · 2023-11-23
> > > **Author Response to Reviewer wGaZ**
> > >
> > > Dear reviewer wGaZ,
> > >
> > > We would like to thank you once again for your valuable time and constructive suggestions, which make our paper stronger.
> > > We have submitted a revised paper integrating all the results and discussion of your suggestions.
> > > And thank you for raising your score. It is a great affirmation for us.
> > >
> > > Best,
> > >
> > > Authors

---

### Official Review · Reviewer_5JXS · 2023-10-29

**Soundness:** 2 fair
**Presentation:** 2 fair
**Contribution:** 2 fair
**Rating:** 6
**Confidence:** 4

**Summary:**

This paper proposes to enhance the efficiency of self-supervised learning (SSL). Based on contrastive SSL methods, such as SimCLR and SimSiam, this paper proposes to reuse and remove the similar regions so as to save computation. To achieve this, this paper first identifies the similarities between regions. However, directly operating on regions would face the region shrinking problem caused by convolution layers, this paper proposes to expand the size of removed region. Compute savings in FLOPs in observed in ImageNet benchmarks.

**Strengths:**

+ Self-supervised learning is computation expensive, this paper proposes to reduce the pretraining cost while preserving the accuracy, which is important topic for the community.

+ The idea of reusing and replacing similar regions is intuitive. Also I am not sure if there are other similar works proposing similar ideas, it is good to see these simple yet effective training techniques.

**Weaknesses:**

- This paper claims that the proposed method is efficient regrading the FLOPs. However, reduced FLOPs may not directly lead to time saving given that the proposed method requires dedicated sparse computation of convolutional kernel. It is important to report the real run time saving to claim efficiency.

- In the title, authors claim the proposed method is generic. It is worth to apply SimWnW to self-supervised vision transformers as well. Moreover, the reuse and replace strategies are expected to be applicable to ViTs since there would be no region shrinking problem in ViTs.

**Questions:**

See weakness.

**Details Of Ethics Concerns:**

Not applicable.

---

> ### Author Response · Authors · 2023-11-21
> **Author Response to Reviewer 5JXS (Part 1/2)**
>
> **We would like to thank the reviewer for the valuable feedback. We also expect this simple yet effective approach could contribute to the community. We have added more results as suggested by the reviewer, including the experimental results on the training time reduction and the experimental results on vision transformers. We hope our response can help clarify the reviewer's questions.**
>
> ---
>
> ### **Q1. Experimental results on training time reduction.**
>
> Thank you for your insightful suggestion.
>
> We added more discussion (Appendix D) and results regarding the actual training time reduction in the revised paper. We have added the actual training time reduction to Tables 1, 2, 3, 4, and 5 in the revised paper. Our SimWnW reduces training time by an average of 24% (up to 35%) for CNN models and 28% for ViT. We measure the actual end-to-end training time under the PyTorch framework using Nvidia Tesla P100 GPU.
>
> Specifically, the FLOPs reduction of our SimWnW mainly comes from two aspects.
>
> For the first aspect, our SimWnW can improve the model convergency speed, indicating a fewer number of training iterations/epochs required to achieve a target accuracy. **This can directly lead to FLOPs reduction and training time saving, which does NOT require any dedicated sparse computation support.** Specifically, SimWnW removes the less important regions, resulting in removing irrelevant features that slow down the learning process, thereby improving model convergence speed.
>
> The second aspect of FLOPs reduction is achieved by removing similar blocks.
>
> For the ViT-based models, removing similar blocks can be directly achieved by removing similar input tokens, resulting in a reduced input sequence length. **This can also directly achieve acceleration, while does NOT require any dedicated sparse computation support.** For the case of using CNN models, SimWnW indeed requires some support for sparse computation. This is a similar problem faced by the designs in other fields, such as sparse training or weight pruning. This usually can be solved in different ways.
>
> For general-purpose devices such as GPUs or mobile devices, SimWnW can be supported by using sparse computation libraries and compiler optimizations.
> For FPGA platforms, the convolution kernels need to be divided into tiles and computed separately. So, the tiling size used in FPGAs can be aligned with the block size used in the SimWnW. In this way, we can easily skip the computation clock cycle for the corresponding block, leading to direct time-saving.
>
> It is worth mentioning that, in our SimWnW, we remove the entire similar blocks during the computation, which creates a coarse-grained sparsity. Compared to the unstructured or irregular sparsity which is usually used in sparse training or weight pruning works, the coarse-grained sparsity created in our SimWnW is much more friendly for sparse computation acceleration on both general-purpose devices and FPGA platforms.
>
> We put Tables R.6 and R.7 (corresponding to Tables 1 and 2 in the revised paper) here for your reference for the training time reduction results. Please also refer to Table R.8 in the answer for Question 2 for more results.
>
>
> >**Table R.6: Results of training from scratch experiments. The encoder for ImageNet is ResNet50 and the encoder for CIFAR is ResNet18. The model is trained for 800 epochs.**
> | **Method**            	| **ImageNet** |  	|  	| **CIFAR-10** |	|  	| **CIFAR-100** |   |  	|
> |-------------|----------|--------|---------------|--------------|------------|----------------|--------------|------------|---------|
> |                                     	| Acc.                     	| Training FLOPs           	| Training Time      	| Acc.     	| Training FLOPs | Training Time | Acc.     	| Training FLOPs | Training Time |
> | SimCLR                              	| 66.39                    	| 100\%                    	| 100\%              	| 90.16    	| 100\%      	| 100\%     	| 57.34    	| 100\%      	| 100\%     	|
> | SimCLR (Same FLOPs as Ours)	| 64.76             	| 80\%              	| -                 	| 86.07 | 48\%	| -        	| 50.46 | 52\%	| -        	|
> |  **SimCLR + SimWnW (Ours)**  | **66.25**                    	| **80\%**                     	| **89\%**               	| **90.10**    	| **48\%**       	| **68\%**      	| **57.69**    	| **52\%**       	| **73\%**      	|
> | SimSiam                             	| 71.12                    	| 100\%                    	| 100\%              	| 90.80    	| 100\%      	| 100\%     	| 57.21    	| 100\%      	| 100\%     	|
> | SimSiam (Same FLOPs as Ours)   | 69.10             	| 81\%              	| -                 	| 87.28 | 46\%	| -        	| 52.18 | 53\%	| -        	|
> |  **SimSiam + SimWnW (Ours)** | **71.28**                    	| **81\%**                     	| **90\%**               	| **91.17**    	| **46\%**       	| **65\%**     	| **57.94**    	| **53\%**       	| **71\%**      	|

---

> ### Author Response · Authors · 2023-11-21
> **Author Response to Reviewer 5JXS (Part 2/2)**
>
> ### **Q1. Experimental results on training time reduction. (cont.)**
>
> >**Table R.7: Results of transfer learning experiments. The encoder is ResNet50, which is pre-trained on ImageNet and then trained on the three datasets for 100 epochs using self-supervised learning methods. After that, we apply linear evaluation protocol to get the accuracy results.**
> | **Method**                          	| **Stanford Cars** |        	|              	| **FGCV Aircraft** |        	|              	| **CUB-200**  |            |              	|
> |-----------------------------------------|-------------------|----------------|----------------------|-------------------|----------------|----------------------|--------------|----------------|----------------------|
> |                                     	| Acc.          	| Training FLOPs | Training Time | Acc.          	| Training FLOPs | Training Time | Acc.         | Training FLOPs | Training Time |
> | SimCLR                              	| 46.12         	| 100\%      	| 100\%     	| 48.43         	| 100\%      	| 100\%     	| 35.78    	| 100\%      	| 100\%     	|
> | SimCLR (Same FLOPs as Ours)	| 43.03      | 53\%	| -        	| 46.74  	| 51\%    | -        	| 33.07 | 55\%	| -        	|
> | **SimCLR + SimWnW (Ours)**  | **46.38**         	| **53\%**       	| **79\%**      	| **48.26**         	| **51\%**       	| **75\%**      	| **36.15**    	| **55\%**       	| **84\%**      	|
> | SimSiam                             	| 50.87         	| 100\%      	| 100\%     	| 51.82         	| 100\%      	| 100\%     	| 38.40    	| 100\%      	| 100\%     	|
> | SimSiam (Same FLOPs as Ours)   | 46.22      | 50\%	| -        	| 48.17  	| 55\%    | -        	| 36.80 | 49\%	| -        	|
> | **SimSiam + SimWnW (Ours)** | **50.95**         	| **50\%**       	| **78\%**      	| **51.76**         	| **55\%**       	| **73\%**      	| **38.22**    	| **49\%**       	| **74\%**      	|
>
>
>
>
> ---
>
>
> ### **Q2. In the title, authors claim the proposed method is generic. It is worth to apply SimWnW to self-supervised vision transformers as well. Moreover, the reuse and replace strategies are expected to be applicable to ViTs since there would be no region shrinking problem in ViTs.**
>
>
> Thank you for your valuable suggestion.
>
> There is indeed no removed region shrinking problem in ViTs, which makes it easier to apply SimWnW to the ViTs. We also do not need to consider the block size since ViTs naturally divide the image into many tokens, typically in the size of 16x16. Therefore, removing similar blocks can be directly achieved by removing similar input tokens, resulting in a reduced input sequence length.
>
> We use a well-recognized self-supervised vision transformer learning framework DINO [1] as our baseline. As shown in Table R.8, SimWnW significantly reduces the training cost by 38\% without accuracy loss compared to the DINO baseline.
>
> We also compare our approach to an efficient vision transformer training framework EViT [2]. For a fair comparison, we apply EViT to DINO and let it consume the same FLOPs as our approach. As shown in Table R.8, SimWnW achieves 0.82\% higher accuracy than EViT with the same training computation.
>
> >**Table R.8: Evaluation on Vision Transformers. The encoder is DeiT-S and the dataset is ImageNet.**
> | **Method**                 	| **Acc.** | **Training FLOPs** | **Training Time** |
> |--------------------------------|----------|--------------------|-------------------|
> | DINO                       	| 58.95\%  | 100\%          	| 100\%         	|
> | DINO + EViT                	| 58.01\%  | 57\%           	| 72\%          	|
> | **DINO + SimWnW**  | **58.83\%**  | **57\%**           	| **72\%**          	|
>
>
>
> We have added the experiments on ViTs in Appendix A and Table 5 in the revised paper.
>
>
> [1] Caron, Mathilde, et al. "Emerging properties in self-supervised vision transformers." Proceedings of the IEEE/CVF international conference on computer vision. 2021.
>
> [2] Liang, Youwei, et al. "Not all patches are what you need: Expediting Vision Transformers via Token Reorganizations." International Conference on Learning Representations. 2022.

---

> ### Author Response · Authors · 2023-11-22
> **Author Response to Reviewer 5JXS**
>
> Dear Reviewer 5JXS,
>
> Thanks for your time and reviewing efforts! We appreciate your valuable comments.
>
> We provide suggested results in the authors' response, including the data and discussion on training time reduction, and experimental results on vision transformers. We hope our responses have answered your questions. It would be our great pleasure if you would consider updating your review or score.
>
> Best,
>
> Authors

---

> > ### Comment · Reviewer_5JXS · 2023-11-22
> > **Good rebuttal, raise my score from 5 to 6**
> >
> > Hi, Thanks for your additional results. They've solved my questions and I've raised my score from 5 to 6.

---

> ### Author Response · Authors · 2023-11-22
> **Author Response to Reviewer 5JXS**
>
> Thank you for raising the score and your time spent reviewing our paper. This is a great affirmation of our work. Your comments are very constructive (e.g., providing actual training time reduction results and applying our method to vision transformers to show generalizability), which makes our paper stronger. We have addressed all your comments in our revision. Thank you again for your valuable time.
>
> Best,
>
> Author

---

### Official Review · Reviewer_Aoaq · 2023-10-30

**Soundness:** 2 fair
**Presentation:** 3 good
**Contribution:** 2 fair
**Rating:** 6
**Confidence:** 4

**Summary:**

The authors aim to improve the training efficiency of self-supervised learning (SSL) and they propose a similarity-based SSL framework called SIMWNW. SIMWNW removes less important regions (remove most similar regions in two views) in augmented images and feature maps and saves the training cost. Experimental results show that SIMWNW reduces the amount of computation costs in SSL.

**Strengths:**

1. This paper is well-written and easy to follow.
2. The authors analyze the importance of different regions on augmented images by removing and reusing similar blocks for the two branches.
3. The authors show that the removed region will shrink after convolution operation and they propose to expand the size of removed region in the feature map.
4. Experimental results show that the proposed method can achieve comparable accuracy using fewer training FLOPs.

**Weaknesses:**

1. Compared with the training FLOPs, the actual time used for training is more important, and the authors did not report it. How much the proposed method can reduce the training time is what we are concerned about. Steps such as matching in the method cannot actually be reflected intuitively through FLOPs.
2. In Table1 and Table2, the authors should list the accuracy of the baseline methods using the same training overhead. For example, how much lower will simclr be than the proposed method when using 80% overhead?
3. Do the training FLOPs in Table2 refer to pre-training or downstream fine-tuning? If it is the former, why is it different from Table1？If it is the latter, how is the proposed method used in single-branch supervised learning?
4. From Figure 6, I cannot see the obvious advantages of the proposed method. I suggest the author change the horizontal axis to training hours.
5. Some related works [1], [2].

[1] Fast-MoCo: Boost Momentum-based Contrastive Learning with Combinatorial Patches. ECCV2022.

[2] Rethinking Self-Supervised Learning: Small is Beautiful. arXiv 2103.13559.

**Questions:**

Please refer to the weaknesses.

---

> ### Author Response · Authors · 2023-11-21
> **Author Response to Reviewer Aoaq (Part 1/3)**
>
> **We appreciate the valuable comments from the reviewer. We carefully address all the reviewer’s questions and revise the paper accordingly. We hope our response can help alleviate the reviewer's concern.**
>
> ---
>
> ### **Q1. Experimental results on training time reduction.**
>
> Thanks for your insightful review.
>
> We have added the actual training time reduction to Tables 1, 2, 3, 4, and 5 in the revised paper. We measure the end-to-end training time under the PyTorch framework using Nvidia Tesla P100 GPU. Specifically, our SimWnW reduces training time by an average of 24% (up to 35%) for CNN models and 28% for ViT.
>
> Please also refer to Table R.2 and Table R.3 in the answer to Question 2 for details.
>
> We have also provided a detailed discussion of the training time reduction and FLOPs reduction brought by our approach in the latest response "Author Response on discussions of training time and FLOPs reduction" below.
>
> We have added this discussion to Appendix D in the revised paper.
>
> ---
>
> ### **Q2. Experimental results when SimCLR and SimSiam baselines consume the same training overhead as our proposed method.**
>
> Thank you for the suggestion.
>
> We have added the results when SimCLR and SimSiam baselines consume the same training overhead with our proposed method in Table 1 and Table 2 in the revised paper. As shown in Table R.2 and Table R.3 (Tables 1 and 2 in the paper), SimWnW consistently provides significantly higher accuracy, 3.5\% on average (up to 7.23\%) compared to the baseline consuming the same training overhead. This further strengthens the effectiveness of our proposed method.
>
> >**Table R.2: Results of training from scratch experiments. The encoder for ImageNet is ResNet50 and the encoder for CIFAR is ResNet18. The model is trained for 800 epochs.**
> | **Method**            	| **ImageNet** |  	|  	| **CIFAR-10** |	|  	| **CIFAR-100** |   |  	|
> |----|-----|-----|----|-----|-----|-----|---|-----|---|
> |                                     	| Acc.                     	| Training FLOPs           	| Training Time      	| Acc.     	| Training FLOPs | Training Time | Acc.     	| Training FLOPs | Training Time |
> | SimCLR                              	| 66.39                    	| 100\%                    	| 100\%              	| 90.16    	| 100\%      	| 100\%     	| 57.34    	| 100\%      	| 100\%     	|
> | SimCLR (Same FLOPs as Ours)	| 64.76             	| 80\%              	| -                 	| 86.07 | 48\%	| -        	| 50.46 | 52\%	| -        	|
> |  **SimCLR + SimWnW (Ours)**  | **66.25**                    	| **80\%**                     	| **89\%**               	| **90.10**    	| **48\%**       	| **68\%**      	| **57.69**    	| **52\%**       	| **73\%**      	|
> | SimSiam                             	| 71.12                    	| 100\%                    	| 100\%              	| 90.80    	| 100\%      	| 100\%     	| 57.21    	| 100\%      	| 100\%     	|
> | SimSiam (Same FLOPs as Ours)   | 69.10             	| 81\%              	| -                 	| 87.28 | 46\%	| -        	| 52.18 | 53\%	| -        	|
> |  **SimSiam + SimWnW (Ours)** | **71.28**                    	| **81\%**                     	| **90\%**               	| **91.17**    	| **46\%**       	| **65\%**     	| **57.94**    	| **53\%**       	| **71\%**      	|
>
>
>
> >**Table R.3: Results of transfer learning experiments. The encoder is ResNet50, which is pre-trained on ImageNet and then trained on the three datasets for 100 epochs using self-supervised learning methods. After that, we apply linear evaluation protocol to get the accuracy results.**
> | **Method**                          	| **Stanford Cars** |        	|              	| **FGCV Aircraft** |        	|              	| **CUB-200**  |            |              	|
> |-----|------|-----|-----|-----|-----|------|-----|-----|-------|
> |                                     	| Acc.          	| Training FLOPs | Training Time | Acc.          	| Training FLOPs | Training Time | Acc.         | Training FLOPs | Training Time |
> | SimCLR                              	| 46.12         	| 100\%      	| 100\%     	| 48.43         	| 100\%      	| 100\%     	| 35.78    	| 100\%      	| 100\%     	|
> | SimCLR (Same FLOPs as Ours)	| 43.03      | 53\%	| -        	| 46.74  	| 51\%    | -        	| 33.07 | 55\%	| -        	|
> | **SimCLR + SimWnW (Ours)**  | **46.38**         	| **53\%**       	| **79\%**      	| **48.26**         	| **51\%**       	| **75\%**      	| **36.15**    	| **55\%**       	| **84\%**      	|
> | SimSiam                             	| 50.87         	| 100\%      	| 100\%     	| 51.82         	| 100\%      	| 100\%     	| 38.40    	| 100\%      	| 100\%     	|
> | SimSiam (Same FLOPs as Ours)   | 46.22      | 50\%	| -        	| 48.17  	| 55\%    | -        	| 36.80 | 49\%	| -        	|
> | **SimSiam + SimWnW (Ours)** | **50.95**         	| **50\%**       	| **78\%**      	| **51.76**         	| **55\%**       	| **73\%**      	| **38.22**    	| **49\%**       	| **74\%**      	|

---

> ### Author Response · Authors · 2023-11-21
> **Author Response to Reviewer Aoaq (Part 2/3)**
>
> ### **Q3. Do the training FLOPs in Table2 refer to pre-training or downstream fine-tuning? If it is the former, why is it different from Table1？If it is the latter, how is the proposed method used in single-branch supervised learning?**
>
> Thanks for your valuable question and we apologize for the confusion.
>
> In the transfer learning settings, we follow the setup used in prior works [1].
> The backbone network ResNet50 is initialized using ImageNet-pretrained weights. Then the ResNet50 is trained on downstream datasets (Cars, Aircraft, and CUB200) using a **self-supervised learning (SSL) method**. Finally, the backbone network ResNet50 is evaluated using the linear evaluation protocol, which is supervised learning.
>
> In summary, we do not directly transfer the ImageNet-trained ResNet50 to the downstream datasets using the supervised learning method. Instead, we conduct self-supervised learning on the downstream datasets. So, the training FLOPs in Table 2 refer to downstream self-supervised fine-tuning.
> We clarify it in the caption in Table 2 in the revised paper.
>
> **For a more comprehensive comparison, we also add the experimental results of directly fine-tuning the pre-trained model on the downstream task without self-supervised learning on them.** In this experiment, SimWnW is applied to SimCLR and SimSiam baselines during the self-supervised pre-training in ImageNet. After that, we directly transfer the pre-trained ResNet50 model to downstream datasets using supervised learning. As shown in Table R.4, applying SimWnW during the pre-training stage will not hurt the accuracy.
>
> We have added this experiment in Appendix C and Table 8 in the revised paper.
>
> >**Table R.4: Accuracy results of transferability experiment.**
> | **Method**                          	| **Stanford Cars** | **FGCV Aircraft** | **CUB-200** |
> |-----------------------------------------|-------------------|-------------------|-------------|
> | SimCLR                            	| 38.49         	| 40.46         	| 30.10   	|
> | **SimCLR + SimWnW (Ours)** | **39.90**         	| **41.28**         	| **30.19**   	|
> | SimSiam                             	| 39.78         	| 38.37         	| 32.89   	|
> | **SimSiam + SimWnW (Ours)** | **40.46**         	| **38.92**         	| **33.66**   	|
>
> [1] Shu, Yangyang, Anton van den Hengel, and Lingqiao Liu. "Learning Common Rationale to Improve Self-Supervised Representation for Fine-Grained Visual Recognition Problems." Proceedings of the IEEE/CVF Conference on Computer Vision and Pattern Recognition. 2023.
>
>
> ---
>
>
> ### **Q4. From Figure 6, I cannot see the obvious advantages of the proposed method. I suggest the author change the horizontal axis to training hours.**
>
> Thanks for your constructive suggestion. We have modified the horizontal axis of Figure 6 to training hours in the revised paper. Indeed, this can further demonstrate the advantages of our method.

---

> ### Author Response · Authors · 2023-11-21
> **Author Response to Reviewer Aoaq (Part 3/3)**
>
> ### **Q5. Some related works [1], [2].**
>
> **[1] Fast-MoCo: Boost Momentum-based Contrastive Learning with Combinatorial Patches. ECCV2022.**
>
> **[2] Rethinking Self-Supervised Learning: Small is Beautiful. arXiv 2103.13559.**
>
> Thanks for the valuable review.
>
> These two works are complementary to SimWnW. And SimWnW can further enhance the performance of these two works. As shown in Table R.5, integrating SimWnW to the Fast-MoCo [1] and S3L [2] can further reduce the training cost by 23\% and 30\% without accuracy loss, respectively, demonstrating the compatibility of SimWnW.
>
> >**Table R.5: Compatibility of SimWnW with SOTA efficient SSL framework. The encoder is ResNet50 and the dataset is ImageNet.**
> | **Method**                      	| **Acc.** | **Training Time** |
> |-------------------------------------|----------|-------------------|
> | MoCo v2                         	| 71.12	| 100\%         	|
> | S3L (MoCo v2 based)                            	| 69.96	| 65\%          	|
> | **S3L + SimWnW**   	| **70.06**	| **46\%**          	|
> | MoCo v3                         	| 72.28	| 100\%         	|
> | Fast-MoCo (MoCo v3 based)                   	| 72.46	| 30\%          	|
> | **Fast-MoCo + SimWnW** | **72.30**	| **23\%**          	|
>
>
> Specifically, Fast-MoCo accelerates the training process by adding more positive pairs to regulate the training process, thereby accelerating convergence. S3L accelerates the training process by using smaller-resolution images and a partial backbone network. On the other hand, SimWnW accelerates the SSL in a different dimension, which removes less important image blocks during training.
>
> We integrate our proposed method SimWnW into these two frameworks. We follow the setting in their paper and use the MoCo v2 framework as the baseline for S3L and MoCo v3 as the baseline for Fast-MoCo. Specifically, Fast-MoCo divides the input image in the online branch into four patches and then combines their four output embeddings to form six new embeddings, each of which involves two patches. In this case, the number of positive pairs is six times as normal training. Thus, it can get more supervision signals in each iteration and thus achieves promising performance with fewer iterations. For S3L, we follow their original setting for the ImageNet experiment in their paper, which uses 52x52 input images to train the model for 800 epochs and then uses 224x224 input images to train the model for 200 epochs.
>
> We have added these experimental results in Section 5.4 and Table 4 in the revised paper.

---

> ### Author Response · Authors · 2023-11-22
> **Author Response to Reviewer Aoaq**
>
> Dear Reviewer Aoaq,
>
> Thanks for your time and reviewing efforts! We appreciate your constructive comments.
>
> We provide suggested results in the authors' response, such as the training time reduction, more comparison with baselines, updates on figures, clarification and more experiments on transfer learning experiments, and experiments concerning other related works.
>
> We hope our responses have answered your questions. It would be our great pleasure if you would consider updating your review or score.
>
> Best,
>
> Authors

---

> ### Author Response · Authors · 2023-11-23
> **Author Response on discussions of training time and FLOPs reduction**
>
> ### **Q.1 Experimental results on training time reduction. (Cont.)**
>
> **Here we provide more discussions about the FLOPs and training time reduction brought by our approach.**
>
> The FLOPs reduction of our SimWnW mainly comes from two aspects.
>
> For the first aspect, our SimWnW can improve the model convergency speed, indicating a fewer number of training iterations/epochs required to achieve a target accuracy. **This can directly lead to FLOPs reduction and training time saving, which does NOT require any dedicated sparse computation support.** Specifically, SimWnW removes the less important regions, resulting in removing irrelevant features that slow down the learning process, thereby improving model convergence speed.
>
> The second aspect of FLOPs reduction is achieved by removing similar blocks.
>
> For the ViT-based models, removing similar blocks can be directly achieved by removing similar input tokens, resulting in a reduced input sequence length. **This can also directly achieve acceleration, while does NOT require any dedicated sparse computation support.** For the case of using CNN models, SimWnW indeed requires some support for sparse computation. This is a similar problem faced by the designs in other fields, such as sparse training or weight pruning. This usually can be solved in different ways.
>
> For general-purpose devices such as GPUs or mobile devices, SimWnW can be supported by using sparse computation libraries and compiler optimizations.
> For FPGA platforms, the convolution kernels need to be divided into tiles and computed separately. So, the tiling size used in FPGAs can be aligned with the block size used in the SimWnW. In this way, we can easily skip the computation clock cycle for the corresponding block, leading to direct time-saving.
>
> It is worth mentioning that, in our SimWnW, we remove the entire similar blocks during the computation, which creates a coarse-grained sparsity. Compared to the unstructured or irregular sparsity which is usually used in sparse training or weight pruning works, the coarse-grained sparsity created in our SimWnW is much more friendly for sparse computation acceleration on both general-purpose devices and FPGA platforms.

---

### Official Review · Reviewer_vumm · 2023-11-01

**Soundness:** 3 good
**Presentation:** 3 good
**Contribution:** 3 good
**Rating:** 8
**Confidence:** 3

**Summary:**

The authors propose a method for improving the efficiency of SSL methods by discarding features in augmented images and feature maps that are deemed less important, saving computation and reducing the risk of slowing the learning process by providing irrelevant features. The authors propose to remove blocks from pairs of augmented images that share high semantic similarity, in order to prevent unnecessary processing of irrelevant information such as image backgrounds. To this end, they provide a method for semantic matching of block pairs in images, their removal, and the treatment of the resulting feature maps throughout the network. Authors show results for training from scratch and transfer learning compared to a number of other SSL methods, in most cases showing barely degraded performance - or even improved performance - at a significantly reduced computational cost.

**Strengths:**

The authors provide a sensible method for improving computational efficiency of SSL methods, one of their main challenges currently. The authors are very thorough in motivating and describing their method, using illustrative examples throughout the paper. Experimental results are impressive, the proposed method shows good performance in its ability to reduce computational cost while retaining model performance.  A very sound paper overall, with good experimental design. Given that the authors spend some time sculpting the manuscript to improve its readability for the rebuttal, I think it represents an interesting and valuable addition to the CVPR proceedings.

**Weaknesses:**

Overall readability of the paper could be improved, I’m having a bit of a hard time understanding some of the specifics of the approach as outlined in 3.1 and 3.2. Specifically, the block matching as outlined in paragraphs 1 and 2 under 3.1 seem to overlap; from my understanding you first search for most similar block pairs (paragraph 1) after which you calculate similarity for all block pairs (paragraph 2)? Why not calculate similarity for all block pairs directly?

Under 4.1, you indicate that, for a given pair of original and augmented image, you divide the first into blocks and loop for a similar block in the paired image. However, instead of performing an exhaustive search over all possible blocks in the augmented image, you narrow the search to “a specific region surrounding a block’s counterpart in the paired augmented image” to ensure semantic consistency. Where does this block’s counterpart come from? Is it simply the same augmentation applied to the block in the original image, i.e. the location of the original block under a flip? In this case, why would the same block in the augmented image not be the most similar block? Semantically, their content is identical is it not? Could you give an intuition as to why you would want to pair image blocks in the same region in the online and target images but not simply pair exact matches under augmentation?

**Questions:**

Could you give a little more explanation for figure 1. In my opinion, the first two paragraphs of 3.1 read a bit confusingly. What is the distinction between the block matching described in the first paragraph and the similarity calculation after the creation of block pairs in the second paragraph? Aren’t they overlapping?

How does computational complexity of the block-matching factor into the overall training complexity? I.e. do the FLOPs listed in tables 1 and 2 contain the overhead for your method? I think this should definitely be taken into account.

---

> ### Author Response · Authors · 2023-11-21
> **Author Response to Reviewer vumm (Part 1/2)**
>
> **We would like to thank the reviewer for the positive feedback and valuable questions. We appreciate the reviewer's acknowledgment that our proposed work is sensible and the experimental results are impressive. We carefully address all the reviewer’s questions and revise the paper accordingly. We hope our response can help clarify the reviewer's questions.**
>
> ---
>
> #### **Q1. The block matching as outlined in paragraphs 1 and 2 under 3.1 seem to overlap; from my understanding you first search for most similar block pairs (paragraph 1) after which you calculate similarity for all block pairs (paragraph 2)? Why not calculate similarity for all block pairs directly? Could you give a little more explanation for figure 1. In my opinion, the first two paragraphs of 3.1 read a bit confusingly. What is the distinction between the block matching described in the first paragraph and the similarity calculation after the creation of block pairs in the second paragraph? Aren’t they overlapping?**
>
> Thanks for your thorough review and we apologize for the confusion.
>
> The process we described In Section 3.1 consists of two steps:
>
> * Step 1 (Paragraph 1): For each image block in the online branch, we do directly calculate the similarity between this image block and all the blocks from the target branch, and find the most similar block on the target branch according to the similarity to form a block pair. If we assume there are 64 image blocks in the online branch, then we do the same thing for all 64 image blocks and obtain 64 image pairs.
>
> * Step 2 (Paragraph 2): Since we cannot remove the entire image (i.e., all 64 blocks), we need to figure out which blocks should be removed. Therefore, we rank/sort them according to the similarity score (obtained in step 1). And we tried to remove the most similar or most dissimilar p% block pairs respectively to see the impact on accuracy.
>
> So, your understanding is correct. There is no need to recalculate the similarity in the second step. We have modified the text of paragraph 2 in Section 3.1 to clarify it.

---

> ### Author Response · Authors · 2023-11-21
> **Author Response to Reviewer vumm (Part 2/2)**
>
> #### **Q2. Under 4.1, you indicate that, for a given pair of original and augmented image, you divide the first into blocks and loop for a similar block in the paired image. However, instead of performing an exhaustive search over all possible blocks in the augmented image, you narrow the search to “a specific region surrounding a block’s counterpart in the paired augmented image” to ensure semantic consistency. Where does this block’s counterpart come from? Is it simply the same augmentation applied to the block in the original image, i.e. the location of the original block under a flip? In this case, why would the same block in the augmented image not be the most similar block? Semantically, their content is identical is it not? Could you give an intuition as to why you would want to pair image blocks in the same region in the online and target images but not simply pair exact matches under augmentation?**
>
> Thank you for your questions.
>
> Intuitively, the most similar block could be at the corresponding location after augmentation such as flip or rotate. However, if we take the **random scaling and cropping augmentation (zoom-in/zoom-out)** into account (which is very critical in contrastive learning), it will cause some trouble. We may no longer be able to find an exact one-to-one correspondence block pair between the online branch and the target branch. This is one of the reasons that we want to search for the most similar block in a small region.
>
> Another reason that we search in a small region (instead of exhaustive search) is to reduce the computation overhead and ensure the semantic consistency of the paired blocks.
>
> For better illustration, we also include Table R.1 here, to show the impact of the search region size on accuracy. We have added the detailed explanation and results to Appendix B and Table 7 in the revised paper.
>
> >**Table R.1: Accuracy when using different search region sizes. The block size is 30x30 and the results are obtained from the transfer learning experiment on the Stanford Cars dataset. The base framework is SimSiam.**
> | **Size of Search Region** | 45x45 | 60x60 | 75x75 |
> |---------------------------|-------|-------|-------|
> | **Accuracy**          	| 50.72 | 50.95 | 50.98 |
>
>
>
> ---
>
> ### **Q3. Do the FLOPs listed in Tables 1 and 2 contain the overhead for your method?**
>
> Yes, all the training FLOPs listed in this paper include the overhead of block-matching. The detail of the overhead calculation for block-matching is presented in Appendix F in the revised paper.

---

> ### Author Response · Authors · 2023-11-22
> **Author Response to Reviewer vumm**
>
> Dear Reviewer vumm,
>
> Thanks for your time and reviewing efforts! We appreciate your constructive comments.
>
> We provide explanations and clarifications of the questions in the review in the author's response. It would be our great pleasure if you would consider updating your review or score.
>
> Best,
>
> Authors

---

### Meta-Review · Area_Chair_GVdg · 2023-12-08

**Metareview:**

The authors of this research paper aim to improve the training efficiency of self-supervised learning (SSL) by proposing a similarity-based SSL framework called SIMWNW. This framework removes less important regions in augmented images and feature maps, which saves on training costs. To achieve this, the authors first identify similarities between regions and then expand the size of the removed region to avoid the region shrinking problem caused by convolution layers. Experimental results show that SIMWNW reduces the amount of computation costs in SSL, as observed in ImageNet benchmarks. While the idea is clear and the empirical evidence supports the claims, the reach of the proposed method is narrow. Indeed, the proposed approach was mainly tested on SimCLR and conclusions are drawn in a not-so-challenging performance regime. At the same time, the flop gains are not sufficiently drastic to justify broad applicability. Nonetheless I think this paper is sound and should be accepted as a poster presentation at ICLR.

**Justification For Why Not Higher Score:**

The proposed approach was mainly tested on SimCLR and conclusions are drawn in a not-so-challenging performance regime. The flop gains are not sufficiently drastic to justify broad applicability.

**Justification For Why Not Lower Score:**

The claims are clear and supported by empirical evidence.

---

### Decision · Program_Chairs · 2024-01-16

Accept (poster)